# Prostate cancer genetic risk and associated aggressive disease in men of African ancestry

Pamela X. Y. Soh [1], Naledi Mmekwa[2], Desiree C. Petersen [3], Kazzem Gheybi[1], Smit van Zyl[4], Jue Jiang [1], Sean M. Patrick[2], Raymond Campbell[5], Weerachai Jaratlerdseri [1], Shingai B. A. Mutambirwa[6], M. S. Riana Bornman [2] & Vanessa M. Hayes [1,2,4,7] ✉

African ancestry is a significant risk factor for prostate cancer and advanced disease. Yet, genetic studies have largely been conducted outside the context of Sub-Saharan Africa, identifying 278 common risk variants contributing to a multiethnic polygenic risk score, with rare variants focused on a panel of roughly 20 pathogenic genes. Based on this knowledge, we are unable to determine polygenic risk or differentiate prostate cancer status interrogating whole genome data for 113 Black South African men. To further assess for potentially functional common and rare variant associations, here we interrogate 247,780 exomic variants for 798 Black South African men using a case *versus* control or aggressive *versus* non-aggressive study design. Notable genes of interest include *HCP5*, *RFX6* and *H3C1* for risk, and *MKI67* and *KLF5* for aggressive disease. Our study highlights the need for further inclusion across the African diaspora to establish African-relevant risk models aimed at reducing prostate cancer health disparities.

Prostate cancer (PCa) is characterised by substantial ancestral disparity[1], which together with significant heritability[2], suggests an inherited genetic contribution. Specifically, men of African ancestry are at greatest risk, with African Americans 1.7 times more likely to receive a diagnosis and 2.1 times more likely to die from PCa than White American men[3]. Within Sub-Saharan Africa, mortality rates are up to 3.2 times greater than global estimates[4]. For southern Africa, PCa incidence rates largely reflect that reported for Northern America, however, mortality rates are 2.7-fold greater (age-standardised mortality rate of 22 per 100,000 men)[4]. Previously, we (and others) have demonstrated that southern Africa is not only home to genetically diverse populations[5], but that Black South African men are at 2.1-fold increased risk for advanced PCa at presentation compared with African Americans (adjusted for age)[6].

Despite this, there is a notable lack of genetic data for populations across Africa.

As of 8 February 2023, the National Human Genome Research Institute-European Bioinformatics Institute (NHGRI-EBI) genome-wide association study (GWAS) Diversity Monitor reports that while Europeans contribute 95.85% to global GWAS, African Americans or Afro-Caribbeans contribute 0.49%, and Sub-Saharan Africans only 0.15%[7]. PCa GWAS research for Sub-Saharan Africa is no different, with further limitations including a focus on specific variants and study power[8,9]. As outlined in Fig. 1A, studies have been restricted to Ghana (474 cases, 458 controls; 2,837,019 variants)[10], Uganda (560 cases, 480 controls; 118 known PCa risk loci and 17,125,421 imputed variants)[11], South Africa (552 cases, 315 controls; 46 known PCa risk alleles)[12], and a single study (MADCaP) that merged data from Ghana, Nigeria, and South Africa

[1]Ancestry and Health Genomics Laboratory, Charles Perkins Centre, School of Medical Sciences, Faculty of Medicine and Health, University of Sydney, Camperdown, NSW 2006, Australia. [2]School of Health Systems and Public Health, University of Pretoria, Pretoria, South Africa. [3]South African Medical Research Council Centre for Tuberculosis Research, Division of Molecular Biology and Human Genetics, Faculty of Medicine and Health Sciences, Stellenbosch University, Cape Town, South Africa. [4]Faculty of Health Sciences, University of Limpopo, Turfloop Campus, South Africa. [5]Phulukisa health Care, Pretoria, South Africa. [6]Department of Urology, Sefako Makgatho Health Science University, Dr George Mukhari Academic Hospital, Medunsa, South Africa. [7]Manchester Cancer Research Centre, University of Manchester, Manchester M20 4GJ, UK. ✉ e-mail: vanessa.hayes@sydney.edu.au

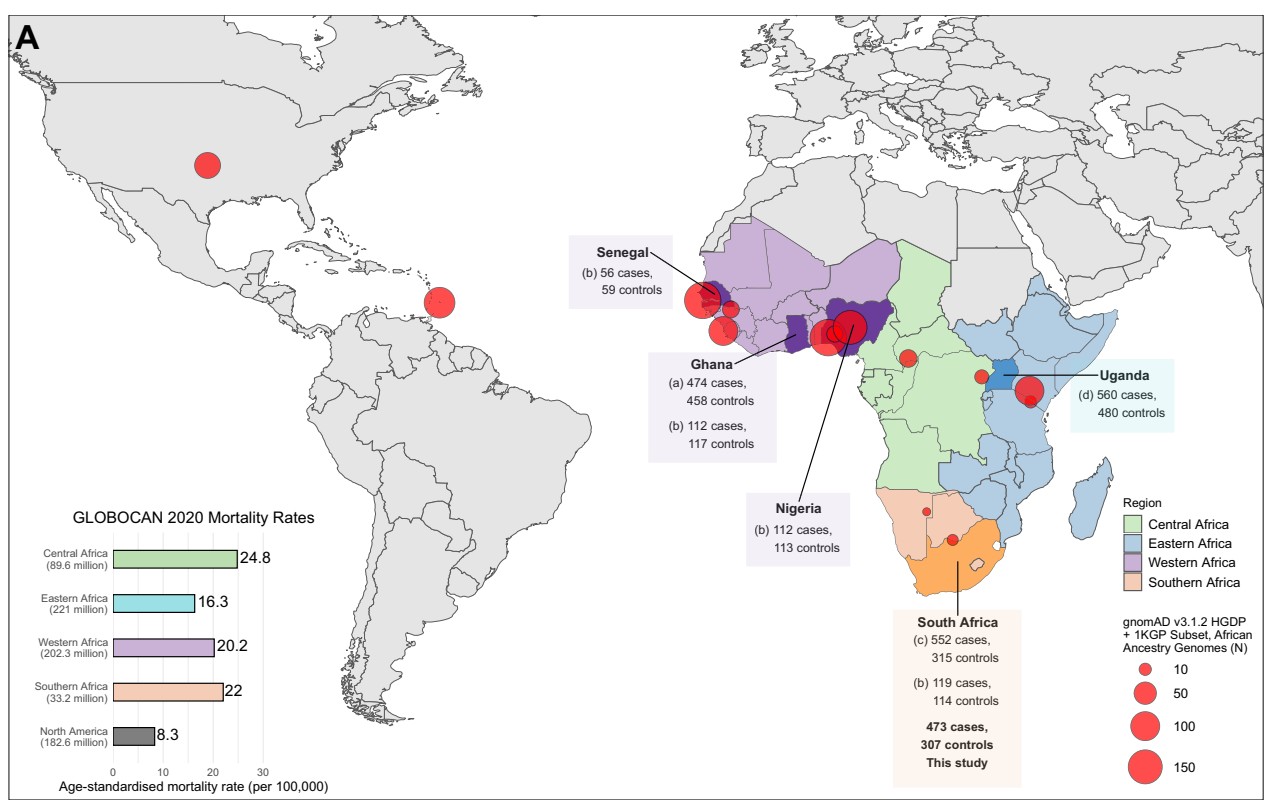

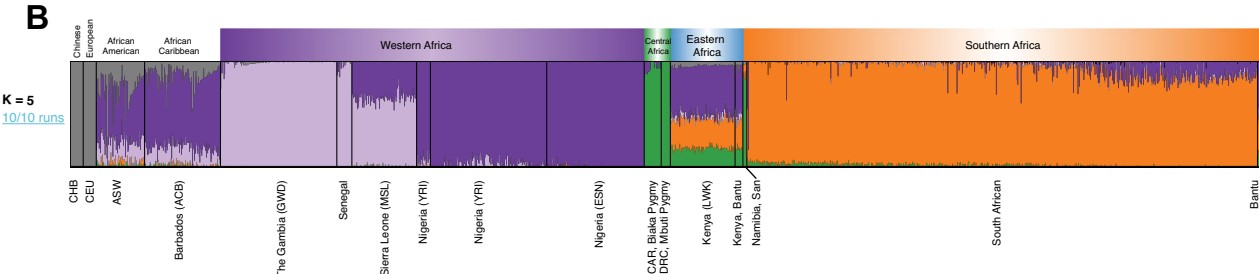

**Fig. 1 | Summary of genome-wide association studies (GWAS) in Africa and ancestral fractions for our dataset. A** Map showing populations across Africa where GWAS studies have been conducted for prostate cancer, and sample locations and sizes of African ancestry from the Human Genome Diversity Project (HGDP) and 1000 Genomes Project (1KGP) subset of gnomAD v3.12[67] (red circles). Mortality rates of prostate cancer from GLOBOCAN 2020 are shown in the bottom left bar plot, with the population size of men in the region indicated in brackets below the region name[4]. Study references: [a]Cook et al.[10], [b]Harlemon et al.[13], [c]Tindall et al.[12], [d]Du et al.[11]. **B** Admixture plot ($K = 5$, cross-validation error = 0.162) which was replicated in 10 out of 10 runs, including 1003 Africans, 20 Europeans, 20 Chinese individuals from the HGDP and 1KGP subset of gnomAD with our dataset of 781 South Africans.

(399 cases, 403 controls; >1.5 million custom markers)[13]. Furthermore, studies associating rare variants with PCa pathogenesis within Sub-Saharan Africa are equally scarce. While a single study from Uganda associated pathogenic *BRCA2, ATM, PALB2* and *NBN* variants with aggressive prostate disease[14], pathogenic variants for *BRCA2, ATM, CHEK2,* and *TP53*, and rare early-onset/familial oncogenic variants of unknown pathogenicity for *BRCA2, FANCA* and *RAD51C*, were linked to advanced disease in Black South African men[15]. The latter study highlights a maximum 30% utility for current European-biased PCa germline testing gene panels for men from southern Africa.

Most recently, the largest meta-analysis of African ancestral PCa GWAS included samples from the Ghana and Uganda, with additional representation from the Democratic Republic of Congo (3,149 cases, 2,547 controls recruited from Africa out of a total of 19,378 cases and 61,620 controls)[16]. The study increased the number of known PCa risk alleles from 269 to 278, and combined to generate a multi-ancestry polygenic risk score (PRS)[16,17]. Demonstrating effective PCa risk stratification for men of African ancestry, including MADCaP validation, African men in the top PRS decile were further distinguished by aggressive disease. Notably and of relevance to this study, of the nine recently identified African-specific/predominant PCa risk variants identified, seven occurred in gene regions, including a protein truncating variant in the prostate-specific gene anoctamin 7 (*ANO7* Ser914Ter), adding a third functionally relevant protein coding variant to the repertoire of known African-specific PCa risk alleles, including previously identified *CHEK2* Ile448Ser[18] and *HOXB13* Ter285Lys[19]. Recognising that only 7.4% of the meta-analysis included men from Sub-Saharan Africa, which in turn represents only a fraction of the region with vast ethno-linguistic and genetic diversity[20], the authors call for further studies across the African diaspora[16].

Placing the Black South African population into global perspective, in this work we use data from 113 whole-genome sequenced men with predominantly high-risk PCa (HRPCa)[21] to evaluate the use of polygenic scoring based on 278 known common risk PCa variants[16].

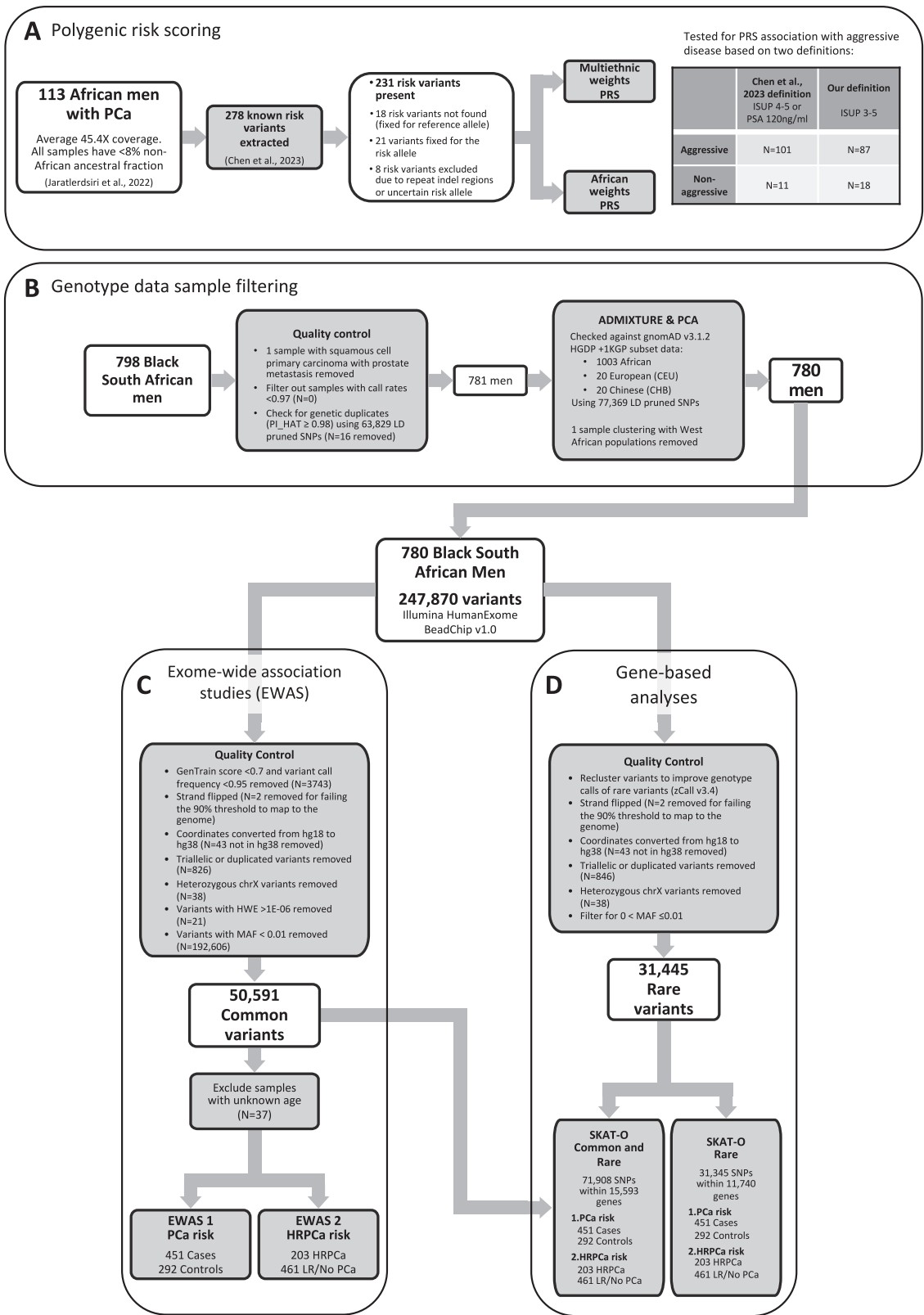

**Fig. 2 | Summary of quality control steps and analyses. A** Summary of the data used for polygenic risk scoring (PRS), (**B**) sample filtering for the genotype data, (**C**) variant filtering for the exome wide association studies (EWAS), and (**D**) rare variant filtering for the gene-based analyses.

To further assess for potentially functional variants associated with PCa and aggressive disease in 798 Black South African men, we conduct an exome-wide association study (EWAS; 247,870 variants), including both common variant (minor allele frequency (MAF) > 0.01) and gene-based rare variant (MAF ≤ 0.01) analyses.

## Results

### Known risk alleles in Black South Africans with PCa

Published whole genome sequenced data generated for 113 Black South African PCa cases (average 45.4X coverage), filtered to represent no more than 8% non-African genetic ancestral contribution (Fig. 2A)[21],

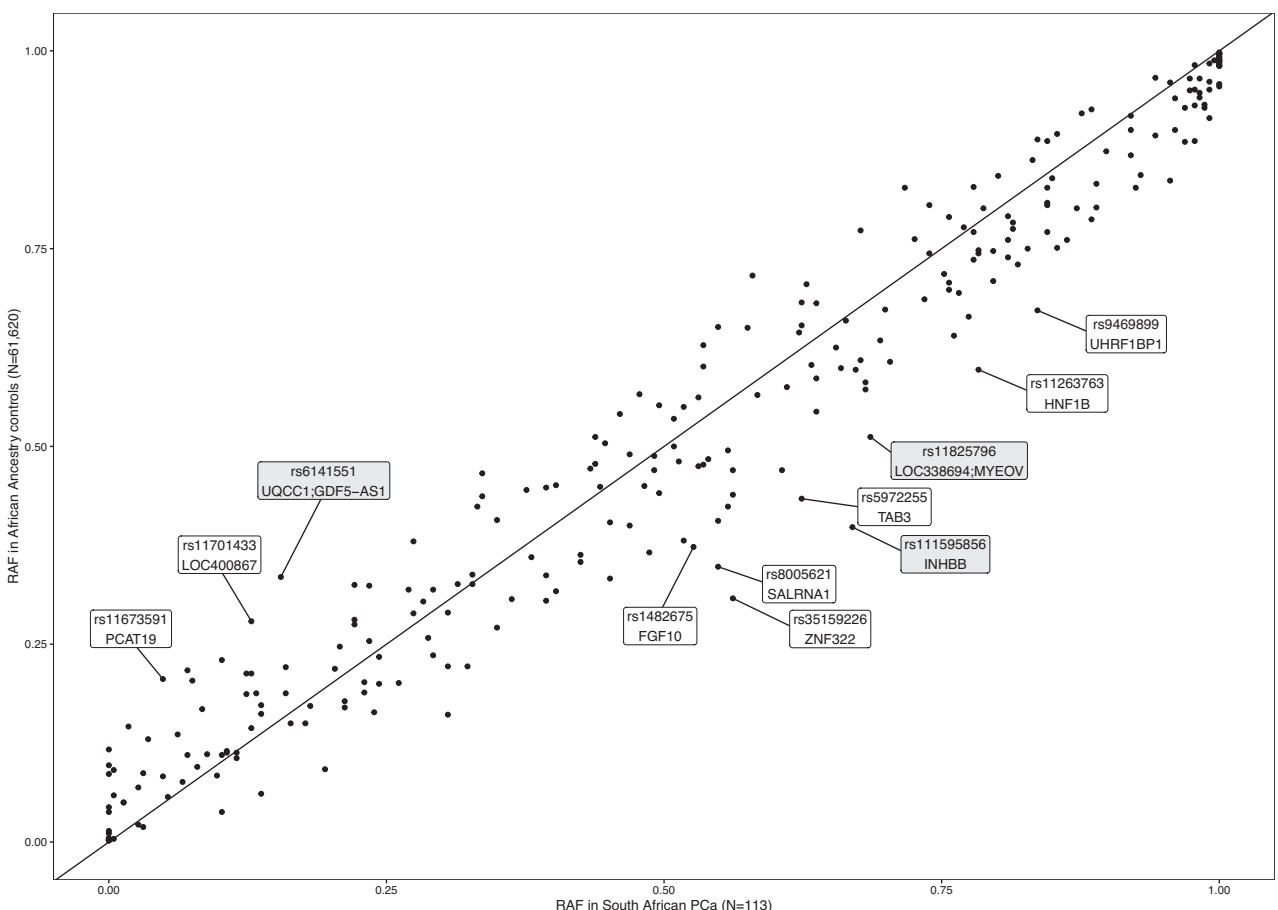

**Fig. 3 | Comparison of differences in risk allele frequencies (RAF).** Comparison of the RAF for 267 out of 278 known risk variants between South African prostate cancer cases (*N* = 113) and African Ancestry controls (*N* = 61,620) as previously reported[16]. Gene labels in white boxes indicate variants that overlap a gene, while gene labels in grey indicate the closest genes to the variant. Note four risk variants had no risk allele frequency reported in the African Ancestry PCa cases[16] and another seven variants in the South African PCa cases were excluded from the plot due to being indel repeats or having unclear risk variants.

were interrogated for known risk variants and previously associated variants from African GWAS. This included the recently reported set of 278 PCa risk variants[16], and the top associated variants from the Ugandan[11] and Ghanaian GWAS[10]. Biased towards HRPCa at presentation, defined as International Society of Urological Pathology (ISUP) grade group ≥3 (Gleason score ≥4 + 3, *n* = 87, 76.99%), the average age of the cohort was 66.9 years (standard deviation (SD) 8.43, range 45–99), with prostate specific antigen (PSA) levels highly elevated (33.63% PSA ≥ 100 ng/ml), as previously observed within the region[6] (Supplementary Data 1). Of the 278 risk variants, 18 (6.47%) were absent, 21 (9.09%) fixed in our Southern African (SA) cohort, and eight were excluded from scoring due to uncertain risk variants (Supplementary Data 2). When compared to previously published risk allele frequencies (RAF) for African-ancestral (AA) controls[16], 11 showed differences in RAF > 0.15, of which eight were more common in our SA population and three were more common in the published largely US-derived AA data (Fig. 3, Supplementary Data 2). The largest differences included rs111595856 (*INHBB*, $RAF_{SA}$ = 0.67, $RAF_{AA}$ = 0.398), rs35159226 (*ZNF322*, $RAF_{SA}$ = 0.562, $RAF_{AA}$ = 0.308), and rs8005621 (*SALRNA1*, $RAF_{SA}$ = 0.549, $RAF_{AA}$ = 0.348). Among the top 136 associated variants in the Ugandan GWAS, three variants showed differences in RAF > 0.15 compared to Ugandan cases and controls, rs6431219 (*BIN1*, $RAF_{SA}$ = 0.416, $RAF_{UGPCS\_Cases}$ = 0.6, $RAF_{UGPCS\_Controls}$ = 0.51), rs61005944 (*ENSG00000237101*, $RAF_{SA}$ = 0.527, $RAF_{UGPCS\_Cases}$ = 0.31, $RAF_{UGPCS\_Controls}$ = 0.22) and rs140698498 (*RBFOX1*, $RAF_{SA}$ = 1, $RAF_{UGPCS\_Cases}$ = 0.87, $RAF_{UGPCS\_Controls}$ = 0.8) (Supplementary Fig. 1). A total of 19 variants were more common in Ugandan controls than in

our SA population (difference in RAF ranged from 0.001 to 0.094, Supplementary Data 3). Among the top 30 associated variants in the Ghanaian GWAS, only one had a difference in RAF > 0.15 (rs28747043, closest gene *MTCO3P1* 18.4 kb away, $RAF_{SA}$ = 0.181, $RAF_{Ghana\_Controls}$ = 0.371; Supplementary Fig. 2). A total of 19 of these variants were more common in Ghanaian controls than our SA cases (difference in RAF ranged from 0.0003 to 0.19; Supplementary Data 4).

**Polygenic risk scores (PRS) in Black South Africans**
The PRS was evaluated using two definitions of aggressive PCa, firstly Chen et al., 2023's definition: ISUP 4-5 or PSA ≥ 20 ng/ml, which grouped our samples into *N* = 101 aggressive and *N* = 11 non-aggressive (one sample with missing PSA and ISUP excluded); and our definition: ISUP 3-5, grouping our samples into *N* = 87 aggressive and *N* = 18 non-aggressive (eight samples with missing ISUP excluded) (Fig. 2). A total of 231 non-fixed variants (Supplementary Data 2) were used to score the SA population with PLINK v1.9[22] using African and multiethnic weights, as previously published[16]. Using Chen et al., 2023's definition of aggressiveness, for African weights, for the aggressive group the mean score was 0.034 (SD = 0.002, range 0.029–0.039) while that of the non-aggressive group was 0.034 (SD = 0.002, range 0.032–0.037). For multiethnic weights, the aggressive group's mean score was 0.041 (SD = 0.002, range 0.035–0.046), and non-aggressive was 0.041 (SD = 0.002, range 0.039–0.045). Using our definition of aggressiveness, for African weights, the aggressive group had a mean score of 0.034 (SD = 0.002, range 0.029–0.039), while the non-aggressive group had a mean score of 0.034 (SD = 0.002, range 0.032–0.037).

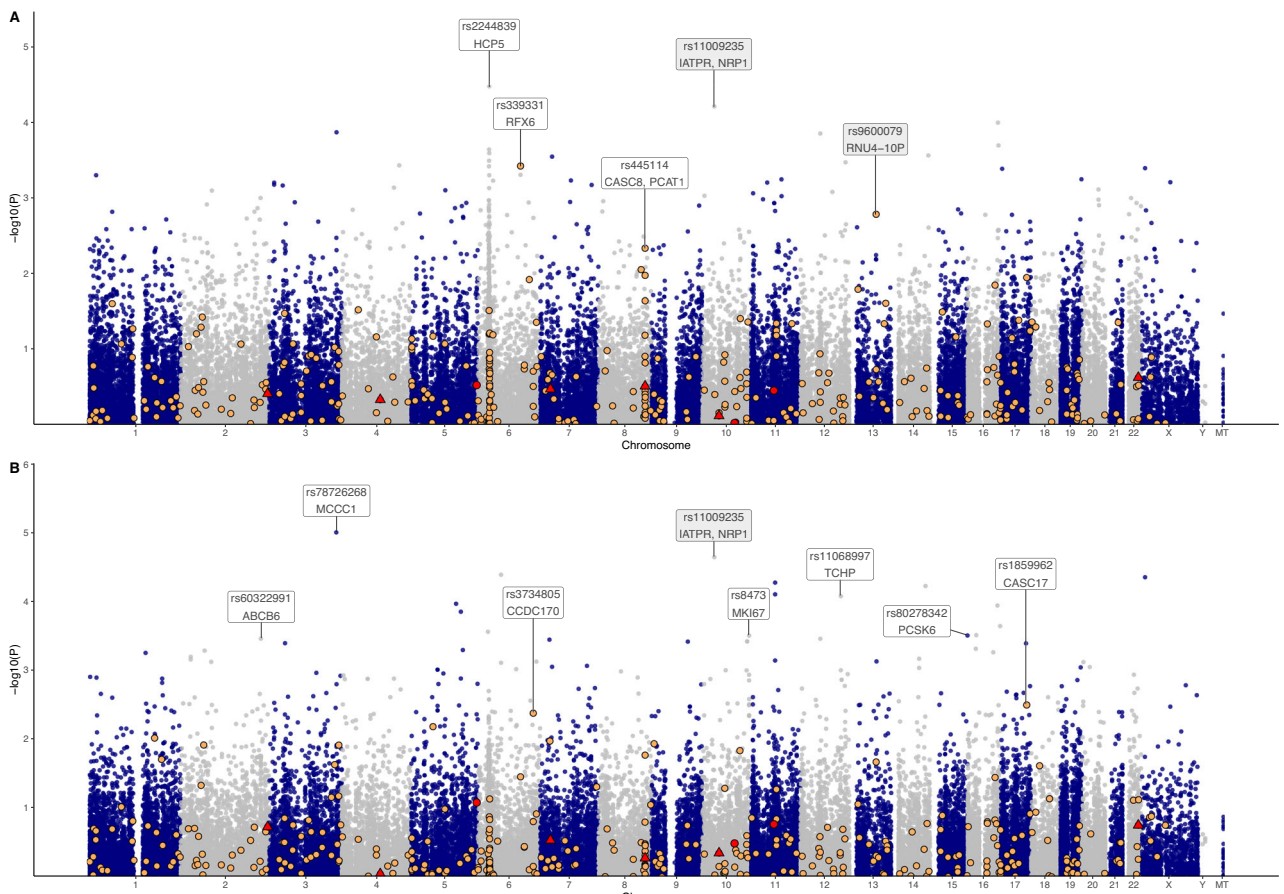

**Fig. 4 | Manhattan plots from the exome-wide association studies (EWAS).**
**A** Manhattan plot of -log$_{10}$ p-values from an age-adjusted logistic regression, with 451 cases and 292 controls. **B** Manhattan plot of -log$_{10}$ p-values from an age-adjusted logistic regression for high grade prostate cancer (ISUP 3-5) cases ($N = 203$) against low risk or no PCa ($N = 540$). Known risk variants ($N = 9$) from the recently described set of 278 variants[16] are shown as red circles, and known cancer variants as summarised previously[13] are shown as orange circles, while variants in both datasets are represented in red triangles. Variants labelled in a white box indicate the overlapping gene, while those labelled in grey are the closest genes.

With multiethnic weights, we observed for the aggressive group a mean 0.041 (SD = 0.02, range 0.035–0.046) and for the non-aggressive group a mean 0.041 (SD = 0.02, range 0.039–0.045) (Supplementary Figs. 3, 4).

No significant associations were detected using either the multiethnic study's definition of aggressiveness[16]: African score OR per SD = 1.38, 95% CI = 0.72–2.66; multiethnic score OR per SD = 1.24, 95% CI = 0.65–2.34; nor using our definition of HRPCa: African score OR per SD = 1.01, 95% CI = 0.6–1.71; multiethnic score OR per SD = 1.13, 95% CI = 0.67–1.9.

**Common risk variants associated with PCa in Black South Africans**

A total of 798 Black South Africans were genotyped on the Infinium HumanExome-12 v1.0 BeadChip array (Illumina, California, United States), screening 247,870 variants. A total of 781 remained after sample quality control (QC) filtering (Fig. 2B). The dataset was assessed for regional ancestral clustering using the Human Genome Diversity Project (HGDP) and 1000 Genomes Project (1KGP) subset of gnomAD v3.1.2, including 1,003 African, 20 European, and 20 Chinese samples. Using principal component analysis (PCA, Supplementary Fig. 5) and ADMIXTURE v1.3.0 for $K = 1$ to 10 with five-fold cross-validation (CV) and 10 replications each (Supplementary Fig. 6), with $K = 5$ generating the lowest CV error at 0.162 (Supplementary Fig. 7), we assessed for within-population substructure. Excluding for a single patient that clustered with Nigerian Yoruba and Esan (West African) populations, we confirm that our cohort represents a distinct southern African

genetic ancestry (Fig. 1B). Recruited from Southern African Prostate Cancer Study (SAPCS) presentative urology clinics within South Africa, cases were defined as presenting with clinicopathologically confirmed PCa ($N = 451$) and controls with no histopathological evidence of cancer (i.e. no Gleason score, $N = 292$), with relatively even distribution across the age-representation (average 70.52, range 49–102 vs average 69.99, range 45-99, respectively). Further clinical characteristics are summarised in Supplementary Data 5. For exome-wide association analysis (EWAS) and gene-based analysis, a total of 37 men with unknown age at diagnosis were excluded, leaving 743 men.

After variant QC, 50,591 common variants (MAF > 0.01) remained for further case-control EWAS (Fig. 2C), with no variants reaching genome-wide significance ($q < 0.05$) (Fig. 4A). The QQ plot and genomic inflation factor ($\lambda = 1.06$) from the $P$-values of the EWAS indicated no population stratification in the data (Supplementary Fig. 8). Of the 17 SNPs with a $P$-value of <5E−04 (Table 1), six (35.3%) were in Chromosome 6 including two intronic variants in the HLA-complex P5 (*HCP5*) lncRNA gene (rs2244839, rs12660382), two within or close to *MUC22* (rs1634718, rs1634725), and one each in *LINC00243* (rs1264362) and *RFX6* (rs339331). The *RFX6* variant rs339331 is a known PCa variant (Fig. 4A), although not included in the set of 278 risk variants, was found to be in strong linkage ($r^2 = 0.95$) with *GPRC6A* rs2274911 (Supplementary Fig. 9). The top-ranked SNPs included rs2244839 ($P = 3.4$E−05; OR 1.63, 95% CI:1.29–2.05) in *HCP5*, rs11009235 ($P = 6.26$E−05; OR 1.6, 95% CI:1.27–2.01) located 15.8 kb upstream and 44.9 kb downstream from *IATPR* and *NRP1*, respectively, and rs3865188 ($P = 1.03$E−04; OR 1.58, 95% CI: 1.25–1.99) 9.9 kb upstream of *CDH13*. Other

**Table 1 | Top associated variants (P < 5E−04) from the age-adjusted logistic regression with all cases (N = 451) and controls (N = 292)**

| CHR | SNP | BP | A1/A2 allele | OR (95% CI) | P-value | MAF Cases | MAF Controls | MAF African ancestry | Overlapped/Closest genes (distance away) | SIFT | PolyPhen | ClinVar |
|---|---|---|---|---|---|---|---|---|---|---|---|---|
| 6 | rs2244839 | 31470591 | A/G | 1.63 (1.29–2.05) | 3.40E−05 | 0.38 | 0.276 | 0.418 | HCP5 | | | |
| 10 | rs11009235 | 33132620 | A/G | 1.6 (1.27–2.01) | 6.26E−05 | 0.399 | 0.298 | 0.412 | IATPR (15.8 kb), NRP1 (44.9 kb) | | | |
| 16 | rs3865188 | 82617112 | T/A | 1.58 (1.25–1.99) | 1.03E−04 | 0.395 | 0.296 | 0.374 | CDH13 (9.9 kb) | | | |
| 3 | rs34759333 | 183758910 | A/C | 2.64 (1.6–4.34) | 1.38E−04 | 0.089 | 0.036 | 0.041 | YEATS2 | Dᵃ | B | B |
| 12 | rs7963300 | 53864157 | A/G | 2.15 (1.45–3.19) | 1.43E−04 | 0.126 | 0.063 | 0.092 | ENSG00000286069 | | | |
| 16 | rs56802364 | 84033418 | C/G | 0.59 (0.45–0.78) | 2.05E−04 | 0.145 | 0.22 | 0.106 | SLC38A8 | D | B | B |
| 6 | rs1634718 | 31005088 | G/A | 0.57 (0.42–0.77) | 2.33E−04 | 0.115 | 0.183 | 0.11 | MUC22 (5.4 kb) | | | |
| 6 | rs12660382 | 31475546 | T/C | 1.69 (1.28–2.25) | 2.60E−04 | 0.226 | 0.149 | 0.168 | HCP5 | | | |
| 14 | rs8019453 | 101444441 | T/C | 0.61 (0.46–0.79) | 2.79E−04 | 0.14 | 0.214 | 0.11 | LINC02314 (126 bp) | | | |
| 7 | rs7792020 | 32409956 | C/T | 1.61 (1.25–2.09) | 2.89E−04 | 0.273 | 0.192 | 0.379 | PDE1C | | | |
| 6 | rs1634725 | 31014231 | A/T | 0.6 (0.45–0.79) | 3.27E−04 | 0.14 | 0.211 | 0.169 | MUC22 | | | |
| 12 | rs1798192 | 122716221 | T/G | 0.69 (0.56–0.85) | 3.43E−04 | 0.414 | 0.514 | 0.493 | HCAR3 | T | B | |
| 4 | rs6845322 | 156962953 | A/G | 0.68 (0.56–0.84) | 3.77E−04 | 0.406 | 0.503 | 0.448 | PDGFC | | | |
| 6 | rs339331 | 116888889 | C/T | 0.64 (0.5–0.82) | 3.84E−04 | 0.172 | 0.252 | 0.238 | RFX6 | | | |
| 6 | rs1264362 | 30808813 | G/C | 0.48 (0.32–0.72) | 3.88E−04 | 0.053 | 0.103 | 0.097 | LINC00243 | | | |
| X | rs2897495 | 10808235 | A/G | 1.78 (1.29–2.45) | 4.10E−04 | 0.397 | 0.271 | 0.437 | MID1 | | | |
| 17 | rs114057260 | 4105807 | C/T | 0.34 (0.19–0.62) | 4.18E−04 | 0.02 | 0.055 | 0.023 | ZZEF1 | Dᵃ | PD | |

African ancestry minor allele frequency (of allele A1) was retrieved from gnomAD v3.1.2.
CHR chromosome, SNP single nucleotide polymorphism, BP base pair (GRCh38), A1 minor allele, A2 major allele, OR odds ratio, CI confidence interval, MAF minor allele frequency, D deleterious, B benign, T tolerated, PD possibly damaging.
ᵃLow confidence.

noteworthy associated variants included rs7963300 ($P = 1.43E-04$; OR 2.15, 95% CI: 1.45–3.19), located within an unknown gene *ENSG00000286069* approximately 74.7 kb upstream from a *HOXC* gene cluster (Supplementary Fig. 10), and the nonsynonymous rs114057260 ($P = 4.18E-04$; OR 0.34, 95% CI 0.19–0.62) in *ZZEF1* which is the only predicted deleterious variant (PDV), defined as variants predicted to be deleterious by SIFT and damaging (or possibly damaging) by PolyPhen.

Only 17 of the 278 known PCa risk variants[16] were captured by the exome array data (Supplementary Data 2), with three SNPs found to be fixed for the risk allele (rs77482050, rs33984059, rs61752561), an additional two almost fixed (rs138708, rs17804499), two were fixed for the reference allele (rs77559646, rs74911261), and one rare (MAF < 0.01 rs76832527) in our SA study population. None of the nine remaining SNPs (MAF 0.015 to 0.49) showed risk association (all $P > 0.25$) (Fig. 4A). A total of 367 out of 2477 known cancer variants, summarised previously[13], were genotyped in the exome array (Supplementary Data 6). Among these, the top associated variants included the *RFX6* SNP rs339331 ($P = 0.0002$), intergenic variant rs9600079 (pseudogene *RNU4-10P* 37.5 kb downstream, closest protein-coding gene is *KLF5* 76 kb upstream, $P = 0.0014$), and *CASC8/PCAT1* variant rs445114 ($P = 0.0047$).

### Common risk variants associated with HRPCa in Black South Africans

Further classification of our study cohort as HRPCa (ISUP ≥ 3, $N = 203$) versus low-risk or no PCa ($N = 461$), again showed no genome-wide significance, while 25 SNPs had $P < 5E-04$ (Fig. 4B, Supplementary Fig. 11), of which three were among the top ranking PCa risk EWAS SNPs (rs11009235, rs2897495, rs7963300). Several of the top SNPs are in genes associated with PCa or PCa processes, including *MKI67* (rs8473), *PCSK6* (rs80278342), *ABCB6* (rs60322991), and *TCHP* (rs11068997); or other cancer-associated processes, including *GGA2* (rs1135045), *H1-5* (rs11970638), and *COL15A1* (rs2075662). The rs8473 SNP in *MKI67* was strongly linked ($r^2 = 0.92$) to rs1063535 in the same gene, and moderately linked ($r^2 = 0.4$ to 0.64) to rs34750407, rs11016071, rs10082391, rs1050767, rs12777740, rs11016076, and rs7095325 (Supplementary Fig. 12). SNPs rs60322991 (*ABCB6*), rs877834 (*NPVF*), rs2075662 (*COL15A1*), and rs77944357 (*ABCA10*) have deleterious and potentially damaging effects based on SIFT and/or PolyPhen yet are benign or are lacking prediction by ClinVar (Table 2). The nine SNPs included in the 278 PCa risk allele panel showed non-significance (all $P > 0.11$), while the top associated known cancer variants were rs1859962 in *CASC17* ($P = 0.003$; OR 1.49, 95% CI: 1.142–1.93) and rs3734805 in *CCDC170* ($P = 0.004$; OR 2.0, 95% CI: 1.244–3.2) (Fig. 4B).

### Genes associated with PCa and HRPCa in Black South African men

Derived from more than 12,000 European-biased sequenced genomes, the exomic array was designed with a focus on protein altering (non-synonymous, slicing and nonsense) variants. These were selected based on a minimum of three observations across two or more datasets, and as such many rare variants have been included, allowing for further gene-based analyses (Fig. 2D). After improving rare variant genotype calls using zCall v3.4[23], gene-based analyses were performed using optimal unified sequence kernel association test (SKAT-O). SKAT-O adjusts for small sample size and retains power when variants in a region are causal and the effect is in a single direction (through burden tests) as well as when variants in a region have bidirectional effects and may contain noncausal variants (through SKAT)[24].

Using this method, genes with rare variants associated with PCa risk (family-wise error rate (FWER) < 0.05; Supplementary Fig. 13, Supplementary Data 7) included *H3C1* (rs199943654, $P = 9.91e-05$), *MBP* (rs61742941, $P = 1.23e-04$), and *MTG1* (3 variants including

predicted deleterious variant (PDV) rs138851534, $P = 1.59e-04$). Although no significance was observed for HRPCa (Supplementary Fig. 14), the top associated gene was *EPS15* (3 rare variants, $P = 1.47e-04$). Through further analyses for common and rare variants (Supplementary Fig. 15, Supplementary Data 8), we show *MBP* to be significantly associated with PCa risk (1 rare and 5 common variants, $P = 1.26e-04$) and included the rare PDV rs61742941. For the HRPCa common and rare variants analysis (Supplementary Fig. 16, Supplementary Data 9), we found *KLF5* to be significantly associated with aggressive disease (1 rare variant, 3 common variants, $P = 1.52e-04$), including the common PDV rs115503899.

### PCa variance explained by exome SNPs

GREML was used to estimate the phenotypic variance explained by genetic variance (SNP heritability). The 49,534 common autosomal SNPs (post-QC) in GREML explained 48.19% (standard error (SE) 22.2%, $P = 4.77E-03$) of disease liability for PCa, and only 17.78% (SE 8.19%) when transformed for PCa prevalence of 0.001 (Table 3). Common and rare autosomal SNPs together ($N = 80,421$) explained 50.7% (SE 25.13%, $P = 0.014$) of disease liability for PCa and at a prevalence of 0.001, explained 18.7% (SE 9.27%).

Using the top 16 SNPs from the EWAS with all cases and controls, only 5.07% (SE 1.78%, $P = 1.59E-25$) of disease liability was explained. When stratifying the cohort by HRPCa *versus* low-risk/no-PCa at an estimated prevalence of 0.0004 for HRPCa, all 49,534 autosomal SNPs explained 16.15% (SE 11.17%, $P = 0.065$) of disease liability, 9.24% (SE 2.31%, $P = 2.3E-44$) of which could be explained using the top 25 SNPs from the HRPCa EWAS.

## Discussion

In this study, motivated by limited studies having identified three African-specific protein-altering risk alleles[19,25], we examined PCa risk and aggressive disease associations and have provided much needed evaluation of known PCa risk alleles within the under-represented region of southern Africa. While dwarfed in sample size compared to European-ancestral or African American GWAS studies, this study highlights significant resources and efforts needed to elucidate the genetic contribution to ancestrally-driven PCa health disparities across the African diaspora.

When examining the previously reported allele frequencies of the 278 known risk variants[16] in our population, large differences in RAF (>0.15) were observed for 11 variants, including those in the genes *INHBB* (rs111595856), *ZNF322* (rs35159226), *SALRNA1* (rs8005621), *FGF10* (rs1482675), *HNF1B* (rs11263763), *PCAT19* (rs11673591), and *TAB3* (rs5972255). These differences could mean that alternate oncogenic pathways or epigenetic regulation are at play in southern Africa. Importantly, although we were restricted in sample size and biased to aggressive PCa, we were unable to replicate previous findings of the multiethnic PRS association with aggressive disease[16], nor with our definition of HRPCa. Notably, the variant rs72725854, which is the most strongly associated risk variant for PCa in men of African ancestry with an allele frequency of 6.1%[26], was present at a frequency of 13.7% in our case-only South African population.

In the classic case-control PCa EWAS analysis, the top associations included variants in *HCP5* and *RFX6*, and near *IATPR* and *NRP1*. The histocompatibility leucocyte antigen (HLA) complex P5 (*HCP5*) is a long non-coding RNA located in the HLA class I region and has shown aberrant expression in multiple cancers, including PCa[27]. A single study investigating *HCP5* in PCa using tissues and cell lines, found high expression of HCP5 to be positively correlated to prostate tumour metastasis[28]. Expression of HCP5 acted as a sponge for miR-4656, preventing miR-4656 from suppressing the cell migration-inducing hyaluronidase 1 (*CEMIP*) gene, leading to upregulated expression of CEMIP, which plays a key role in tumour proliferation[28,29]. Functional experiments will be needed to explore whether the two associated

**Table 2 | Top associated variants (*P* < 5E−04) from the age-adjusted logistic regression with high-risk prostate cancer cases (ISUP grade group 3-5; HRPCa; *N* = 203) against low-risk or no PCa (LR/No PCa; *N* = 461)**

| CHR | SNP | BP | A1/A2 alleles | OR (95% CI) | P-value | MAF HRPCa | MAF LR/No PCa | MAF African ancestry | Overlapped/Closest genes (distance away) | SIFT | PolyPhen | ClinVar |
|---|---|---|---|---|---|---|---|---|---|---|---|---|
| 3 | rs78726268 | 183037313 | T/C | 4.78 (2.39–9.57) | 9.74E-06 | 0.062 | 0.014 | 0.026 | MCCC1 | T | B | B/LB |
| 10 | rs11009235 | 33132620 | A/G | 1.73 (1.34–2.22) | 2.23E-05 | 0.436 | 0.321 | 0.412 | IATPR (15.8 kb), NRP1 (44.9 kb) | | | |
| 6 | rs34644326 | 63712963 | C/G | 3.15 (1.82–5.44) | 4.02E-05 | 0.079 | 0.029 | 0.048 | PHF3 | T | | B |
| X | rs2897495 | 10808235 | A/G | 2.04 (1.45–2.86) | 4.37E-05 | 0.463 | 0.3 | 0.437 | MID1 | | | |
| 11 | rs1194099 | 65582378 | A/T | 1.82 (1.36–2.43) | 5.23E-05 | 0.264 | 0.168 | 0.158 | EHBP1L1 | T[a] | | |
| 14 | rs28670114 | 93622023 | C/T | 1.91 (1.39–2.61) | 5.87E-05 | 0.224 | 0.137 | 0.164 | UNC79 | T[a] | B | |
| 11 | rs75062275 | 65581112 | A/G | 1.79 (1.34–2.39) | 7.75E-05 | 0.261 | 0.168 | 0.154 | EHBP1L1 | T[a] | | |
| 12 | rs11068997 | 109945336 | A/G | 2.05 (1.44–2.94) | 8.25E-05 | 0.16 | 0.089 | 0.094 | TCHP, GIT2 | D[a] | | B |
| 5 | rs12658464 | 122153427 | T/C | 0.61 (0.47–0.78) | 1.06E-04 | 0.291 | 0.403 | 0.543 | ZNF474 | | | |
| 16 | rs61734425 | 81207531 | G/T | 2.55 (1.59–4.11) | 1.13E-04 | 0.096 | 0.045 | 0.036 | PKD1L2 | | | |
| 5 | rs10074434 | 135153492 | G/T | 0.62 (0.49–0.79) | 1.39E-04 | 0.35 | 0.457 | 0.548 | PITX1-AS1 | | | |
| 16 | rs75279347 | 88532007 | A/G | 2.37 (1.5–3.74) | 2.24E-04 | 0.099 | 0.044 | 0.038 | ZFPM1 | T | | |
| 6 | rs11970638 | 27867099 | C/T | 1.91 (1.35–2.71) | 2.72E-04 | 0.158 | 0.09 | 0.088 | H1-5 | T | | |
| 16 | rs1135045 | 23478390 | C/G | 0.64 (0.5–0.81) | 3.04E-04 | 0.399 | 0.505 | 0.416 | GGA2 | T | B | |
| 15 | rs80278342 | 101313285 | A/G | 3.1 (1.68–5.74) | 3.07E-04 | 0.062 | 0.021 | 0.012 | PCSK6 | | | |
| 10 | rs8473 | 128101314 | T/C | 0.64 (0.5–0.82) | 3.09E-04 | 0.416 | 0.521 | 0.534 | MKI67 | T | | |
| 2 | rs60322991 | 219216123 | T/C | 0.57 (0.42–0.78) | 3.42E-04 | 0.153 | 0.241 | 0.153 | ABCB6 | D | PD | B |
| 12 | rs79633300 | 53864157 | A/G | 2 (1.37–2.92) | 3.43E-04 | 0.144 | 0.082 | 0.092 | ENSG00000286069 | | | |
| 7 | rs877834 | 25228315 | C/T | 1.71 (1.28–2.3) | 3.54E-04 | 0.239 | 0.154 | 0.117 | NPVF | D | PD | |
| 10 | rs35299879 | 122841282 | A/C | 2.63 (1.54–4.49) | 3.75E-04 | 0.074 | 0.028 | 0.014 | CUZD1 | T | B | |
| 10 | rs34246902 | 122841318 | C/G | 2.63 (1.54–4.49) | 3.75E-04 | 0.074 | 0.028 | 0.014 | CUZD1 | T | B | |
| 9 | rs2075662 | 98985952 | A/G | 0.59 (0.44–0.79) | 3.78E-04 | 0.182 | 0.274 | 0.312 | COL15A1 | D | P | |
| 3 | rs7648947 | 43132737 | A/G | 0.62 (0.47–0.81) | 3.98E-04 | 0.251 | 0.347 | 0.361 | POMGNT2 (26.7 kb) | | | |
| 17 | rs77944357 | 69215844 | T/G | 2.83 (1.59–5.03) | 4.01E-04 | 0.067 | 0.023 | 0.018 | ABCA10 | D | PrD | |
| 16 | rs9937453 | 22144308 | A/G | 1.55 (1.21–1.99) | 4.80E-04 | 0.51 | 0.413 | 0.53 | VWA3A | T | B | |

African ancestry minor allele frequency (of allele A1) was retrieved from gnomAD v3.1.2.
*CHR* chromosome, *SNP* single nucleotide polymorphism, *BP* base pair (GRCh38), *A1* minor allele, *A2* major allele, *OR* odds ratio, *CI* confidence interval, *MAF* minor allele frequency, *T* tolerated, *B* benign, *LB* likely benign, *D* deleterious, *PD* possibly damaging, *PrD* probably damaging.
[a]Low confidence.

**Table 3 | GREML (genome-based restricted maximum likelihood) analyses results for proportion of phenotypic variance explained by genetic variance**

| GRM | Phenotype | SNPs (N) | Samples (N) | No prevalence specified V(G)/Vp ± SE (%) | At PCa prevalence of 0.001 V(G)/Vp_L ± SE (%) | At HR PCa prevalence of 0.0004 V(G)/Vp_L ± SE (%) | P value |
|---|---|---|---|---|---|---|---|
| Common SNPs | PCa | 49534 | 780[a] | 48.19 ± 22.2 | 17.78 ± 8.19 | | 4.77E-03 |
| Common SNPs | HRPCa | 49534 | 679[a] | 38.45 ± 26.59 | | 16.15 ± 11.17 | 0.065 |
| Common SNPs | ISUP grade group | 49534 | 372 | 1E-06 ± 44.93 | | | 0.5 |
| Common + Rare SNPs | PCa | 80421 | 780[a] | 50.7 ± 25.13 | 18.7 ± 9.27 | | 0.014 |
| Common + Rare SNPs | HRPCa | 80421 | 679[a] | 23.93 ± 25.74 | | 8.8 ± 9.47 | 0.148 |
| Common + Rare SNPs | ISUP grade group | 80421 | 372 | 1E-06 ± 33.84 | | | 0.5 |
| Cases vs controls top SNPs | PCa | 16 | 780[a] | 13.73 ± 4.82 | 5.07 ± 1.78 | | 1.59E-25 |
| HRPCa top SNPs | HRPCa | 24 | 679[a] | 25.13 ± 6.29 | | 9.24 ± 2.31 | 2.30E-44 |
| HRPCa top SNPs | HRPCa among cases | 24 | 372 | 25.01 ± 6.81 | | 7.78 ± 2.12 | 1.34E-22 |
| HRPCa top SNPs | ISUP grade group | 24 | 372 | 17.75 ± 5.9 | | | 3.89E-16 |

Phenotypes included prostate cancer (PCa) diagnosis; high risk prostate cancer (HRPCa), defined as high risk (ISUP 3-5) *versus* low risk (ISUP 1-2) or no PCa (controls); or ISUP grade group (1 to 5). Genetic relationship matrices (GRM) were constructed using all autosomal SNPs that passed EWAS QC (common SNPs), common plus rare autosomal SNPs, or top autosomal SNPs from EWAS analyses (P < 1E−04). V(G)/Vp refers to the ratio of genetic variance (V(G)) over phenotypic variance (Vp), while V(G)/Vp_L indicates V(G)/Vp transformed to the underlying liability scale based on the prevalence supplied.
*SE* standard error.
[a]Includes samples that are lacking age, which were excluded from EWAS analyses.

variants from this study affect expression levels of HCP5. Conversely, the variant located between the lncRNA *IATPR* and *NRP1* provides a potential proxy for a yet unknown gene-associated variant, Notably, *IATPR* has been found to promote cell migration and development in other cancers[30,31], while *NRP1* is an androgen-repressed gene that plays a role in cancer progression with its expression associated with prostate tumour grade and biochemical recurrence[32].

Intriguingly, the member of the regulatory factor X family of transcription factors *RFX6* gene variant rs339331 associated with a decreased PCa risk (C-allele OR 0.64, 95% CI:0.5−0.82, P = 4.6E−04) was in strong linkage with *GPRC6A* rs2274911 ($r^2$ = 0.95). The rs339331 association replicates findings from previous studies of PCa in Ghanaian, Japanese, and Chinese men[10,33,34]. Conversely, the T-allele has been shown to increase HOXB13 binding to a transcriptional enhancer, which upregulates the expression of RFX6 associated with tumour progression, metastasis and biochemical relapse[35], as well as upregulating GPRC6A expression[36]. Additionally, the A-allele at rs2274911 has been associated with increased PSA levels[37]. Representing the major allele in the current study (MAF = 0.7941) could potentially contribute (at least in part) to elevated PSA levels observed for Black South African men, irrespective of PCa status[6]. Notably, the intron variant rs339351 in the *RFX6* gene is among the 278 known PCa risk variants[16], differing in frequency by 0.039 (RAF$_{SA}$ = 0.783, RAF$_{AA}$ = 0.744). As noted for other PCa risk variants, such as European-specific *HOXB13* p.Ile448Ser (rs138213197)[38] and African-specific *HOXB13* p.Ter285Lys (rs77179853)[19,39], it is possible that different ancestral-specific causative variants may represent the same PCa gene.

In the HRPCa EWAS, the top associated variants were in several genes relevant to PCa, including *MKI67*, *PCSK6*, *ABCB6*, and *TCHP*. *MKI67* encodes the Ki67 protein, a widely used diagnostic marker of proliferation in numerous human cancers, with increased expression associated with poor prognosis in localised PCa[40]. One study reported low PSA and high Ki67 expression in patients with HRPCa and *TMPRSS2-ERG* fusion gene, whereas high PSA and low Ki67 expression predominated in patients with low-risk disease and favourable outcomes[41]. Associating the T-allele of *MKI67* rs8473 with reduced odds for HRPCa (P = 3.09E-04, OR = 0.64, 95% CI:0.5−0.82), warrants further investigation into the potential of this allele to reduce Ki67 expression. The *PCSK6* variant rs80278342 is not regarded as a PDV in this study, but an isoform of PCSK6 has previously been identified as a

plasma biomarker for PCa, and expression levels have been correlated with ERG tumour status and ISUP grade group[42]. *ABCB6* codes for an ABC transporter, with expression levels linked to chemoresistance[43]. While increased ABCB6 expression has been reported in PCa, deregulated expression has been further associated with recurrent *versus* non-recurrent disease[44]. Although the correlation between ABCB6 expression levels and PCa grade has not yet been examined, increased ABCB6 has been associated with histological grade in gliomas[45]. The PDV rs11068997, associated with HRPCa in our study, is located within the tumour suppressor gene *TCHP*, shown to inhibit cell growth in PCa cells[46]. Other notable SNPs associated with HRPCa in our study and in genes representing cancer-associated processes include; *GGA2* involved in cell growth[47], *H1-5* with transcriptional regulatory effects[48], and *COL15A1* shown to have tumour suppressive effects[49]. Lastly, the rs7963300 associated SNP upstream of a *HOXC* gene cluster, is of note as increased expression of HOX genes have been observed in prostate tumours[50].

Since the distribution of several disease-associated alleles across a range of African populations has previously shown large variation[51], it is plausible that, on top of potentially different oncogenic pathways or epigenetic regulation, germline PCa risks may also differ between regions within Sub-Saharan Africa. African American bias and within continental representation limited to a snapshot of largely west and east African ancestral diversity[10,11,16,17] provides an explanation for the limited replication of previous GWAS findings in our SA-focused study. This is exemplified in the *HOXB13* risk variant (rs77179853) found in men of West African ancestry being absent in Uganda and South Africa, and possibly arising after the Bantu migration from western to eastern and southern Africa around 1500−4600 years ago[19]. This variant was also absent in our cohort, along with another recently identified African-specific variant *CHEK2* (rs17886163)[18]. Conversely, the *ANO7* variant (rs60985508)[18] was present (RAF$_{SA}$ = 0.363)[52]. Although further investigations are needed, this study highlights some avenues of interest for future germline studies of PCa across southern Africa.

The gene-based analyses showed association between the genes *H3C1*, *MBP*, and *MTG1* with PCa, and *KLF5* with HRPCa. Modifications to the H3 histone plays a key role in epigenetic regulation, and their relevance to PCa and treatment options have been reviewed elsewhere[53,54]. In this study, significant association of *H3C1* to PCa suggests that differences in transcription may exist, and since the variant was only

present in controls it may suggest a protective effect against PCa. The *MBP* gene encodes the myelin basic protein, which is a constituent of the myelin sheath, and has no obvious role in PCa, however, mouse model studies have shown that neural progenitor cells can invade prostate tumours, triggering neurogenesis and promoting tumour growth and metastasis[55]. The *MTG1* gene is involved in the regulation of mitochondrial translation[56] and although it has no known direct role in PCa, mutations in mitochondria have been associated with PCa aggression[57], including in Black South African men[58]. Conversely, *KLF5* is a transcription factor that has been implicated in several cancers with opposing roles (tumour suppressor or oncogenic driver) depending on the context[59]. Expression levels and post-translational modifications of KLF5, specifically acetylation, are of key interest in PCa with therapeutic implications for chemoresistance[60–64]. While there are no PCa risk variants within *KLF5*, two risk SNPs in close proximity include snRNA rs7489409 (65 kb downstream) and lncRNA rs7336001 (344 kb downstream)[16]. A known cancer variant rs9600079 approximately 76 kb downstream from *KLF5* showed slight association ($P = 0.0017$). Located at chromosome 13q22.1, this region is frequently deleted in PCa[65]. Although earlier PCa cell line and xenograft research found that mutations were rare, deletions and down-regulation of *KLF5* was frequent[66]. In our recent study genome profiling SA derived prostate tumours, we describe a molecular taxonomy we call global mutational subtypes (GMSs), identifying a single *KLF5* African-specific predicted cancer driver mutation[21].

Acknowledging our limited sample size, in turn it cannot be ignored that our EWAS is not only comparable to the previous Ghanian and Uganda PCa GWAS[10], but it provides an as yet unmet regionally focused SA alternative for both risk allele validation and discovery across Sub-Saharan Africa (Fig. 1A). While the HGDP and 1KGP subset from the gnomAD v3.1.2 dataset represents limited (20 genomes) representation across SA[67] (Fig. 1A), we acknowledge and appreciate that through projects like H3Africa (www.h3africa.org)[68] additional resources are becoming available, although the pace remains a fraction of the global effort. As with the exomic-array used in this study[69], commercial genotyping arrays have been designed based on variant frequencies heavily skewed towards Europeans, with arrays from MADCaP and H3Africa tailored for African populations, with the MADCaP array specifically designed for PCa research[13,68]. Furthermore, GREML analyses showed that only 49.71% (±22.71%) of the variance in phenotype could be explained by the autosomal genomic variance, indicating that the genomic risk for PCa was not fully captured by the European-biased exome array. Finally, we appreciate that although we attempted to improve rare variant calls using the zCall[23], the software can introduce false positives[70] and as such, we call for caution when interpreting allele frequencies.

While we appreciate our limited sample size, we were unable to replicate previous findings of an association between multiethnic PRS to PCa aggression in our African population. Consequently, we call for further African-relevant whole-genome sequencing and genome-wide interrogation studies for establishing PRS of relevance for Sub-Saharan Africa. In our exome-wide association analyses, we identified several avenues of interest for further investigation, including *HCP5*, *RFX6*, and *H3C1* for PCa, and *MKI67* and *KLF5* for HRPCa. Clearly, significant resources are necessary to elucidate the genomic variants contributing to ethnic disparity in PCa. The global inclusion of southern African data, a region with the most diverse human populations, will benefit not only the design of African-relevant cancer screening panels and further enhance ancestrally focused SNP arrays, but will also be important for accurate multiethnic PRS.

## Methods
### Ethics approvals and recruitment
Written informed consent was obtained from all participants, with study approval granted by the University of Pretoria Faculty of Health Sciences Research Ethics Committee (HREC #43/2010, with US Federal wide assurance FWA00002567 and IRB00002235 IORG0001762). Study participants were recruited at time of biopsy (diagnosis) from participating Southern African Prostate Cancer Study (SAPCS) urology clinics within the Gauteng and Limpopo Provinces of South Africa. The majority of men presented with a urological or associated complaint without a predetermined prostate specific antigen (PSA) test[6]. Men self-identifying ethno-linguistically, by two generations, as Black South Africans, where included in this study; firstly, irrespective of their PCa diagnosis to undergo whole exome genotyping (case-control study, $N = 798$), and secondly, selected for aggressive PCa at presentation and having undergone deep tumour/blood paired whole genome sequencing ($N = 113$), as recently published[21]. Genomic interrogation was performed under approval granted by the St. Vincent's Sydney HREC (#SVH/15/227) and an executed material and data sharing agreement between the University of Pretoria in South Africa and University of Sydney in Australia. While data is shared, all data remains the property of the University of Pretoria, as chair of the SAPCS data sharing committee.

### Interrogation of known PCa risk alleles in SAPCS
We interrogated population-matched whole genome sequenced data for the distribution of the 278 known risk variants[16], and the top associated variants from the Uganda[11] and Ghana[10] GWAS within a cohort of men selected for bias towards aggressive PCa at presentation. Gene annotations were fetched through ANNOVAR from hg38 ensGene (GENCODE v43, last updated from UCSC 2023-02015)[71]. As recently described[21], deep sequenced data (average 45.4X coverage) was generated for 113 Black South African PCa cases, which were filtered to include samples with no more than 8% non-African genetic ancestral contribution (Fig. 2A). A summary of clinical information is available in Supplementary Data 1.

We scored the South African cases via PLINK v1.9[22] using default settings based on their genotypes at 231 out of 278 available risk variants (Fig. 2A) using multiethnic and African ancestry weights[16]. Aggressiveness was defined as previously described (ISUP 4 or 5, or PSA ≥ 20 ng/ml)[16], as well as using an alternative definition (ISUP 3-5), to use as the outcome variable in a logistic regression using the score (African or multiethnic) and age as covariates.

### SAPCS EWAS data generation and genotype filtering
A total of 798 Black South Africans were genotyped on the Infinium HumanExome-12 v1.0 BeadChip array (Illumina, California, United States), assaying 247,870 variants (Fig. 2B–D). Genotypes were called using the Illumina GenomeStudio 2.0 software following previously published guides[70,72]. Briefly, non-pseudoautosomal variants with poor GenTrain scores were manually re-clustered based on visual inspection to improve the accuracy of genotype calls. No samples were removed as all sample call rates were >0.97.

Several quality control steps were conducted to prepare the dataset for exome-wide association analyses (EWAS) (Fig. 2C). Following manual re-clustering, variants that had a poor GenTrain score (<0.7) and poor call frequency (<0.95) were excluded ($N = 3743$). The remaining 244,127 variants were exported to PLINK format. The strand was converted using scripts by William Rayner from the Wellcome Centre for Human Genetics, Oxford website (https://www.well.ox.ac.uk/~wrayner/strand/), in the process removing two single nucleotide polymorphisms (SNPs) that did not reach the required 90% threshold for mapping to the genome. The variants coordinates were converted from hg18 to hg38 using UCSC's webtool liftOver (http://genome.ucsc.edu/cgi-bin/hgLiftOver), and 43 variants not in hg38 were removed. Triallelic or duplicated variants were removed ($N = 826$). Among duplicated variants pairs, the variant with the higher call rate was kept. Heterozygous chromosome X SNPs ($N = 38$) were removed. Variants with MAF < 0.01 ($N = 192,606$; Supplementary Data 10) and 21 SNPs

that failed the Hardy-Weinberg exact test (threshold 1E−6) were removed, leaving 50,591 variants.

## SAPCS EWAS cohort characterisation

After excluding for a single patient presenting with prostate metastasis with squamous cell primary carcinoma (Fig. 2B), the remaining 797 SAPCS EWAS samples were checked for genetic duplicates. Identity-by-descent (IBD) was calculated using the –genome function in PLINK v1.9[22]. Eight pairs of genetic duplicates were identified (PI_HAT ≥ 0.99), and all 16 individuals were removed from further analysis. The maximum PI_HAT in remaining pairs of individuals was 0.29, so no further samples were removed. To check for genetic admixture or non-African genetic fractions, the exome array variants were extracted from published African ancestral genomes ($N = 1003$) as well as 20 randomly selected individuals from European (CEU) and Han Chinese (CHB) populations each from the human genome diversity project (HGDP) and 1000 genome project (1KGP) subset of gnomAD v3.1.2[67]. The extracted data was merged with the data in the current study and pruned for SNPs based on linkage disequilibrium (LD), using a 50 SNP window moving 5 SNPs at a time, at a variance inflation factor of 1.5 (--indep 50 5 1.5) in PLINK v1.9[22]. The remaining 77,372 SNPs were then used for a principal component analysis (PCA) using PLINK v1.9[22] and plotted with 'ggplot2' in RStudio v4.1.1[73]. While no individual showed substantial non-African genetic contribution, a single study participant clustering near the Nigerian Yoruba and Esan west African populations was excluded (Supplementary Fig. 5), leaving a total of 780 samples. The distribution of samples from this study and previous PCa GWAS studies in Africa[10–13] were plotted on a map using the R package 'rnaturalearth'[74].

To further assess for African-specific ancestral fractions, an unsupervised ADMIXTURE v1.3.0 analysis[75] was performed using the same dataset. The ADMIXTURE analysis was conducted for $K = 1$ to 10 with five-fold cross-validation (CV) and 10 replications each. The tool pong v1.5[76] was used to plot ancestry proportions with a greedy approach set to 0.95 (Supplementary Fig. 6). The $K = 5$ ADMIXTURE run produced the lowest cross-validation error at 0.16182 (Supplementary Fig. 7).

As age is a significant PCa risk factor, 37 samples were further excluded from the EWAS for lack of reported age at diagnosis, leaving a total of 451 clinicopathologically confirmed cases (70.52 years SD ± 9.21, range 49-102) and 292 controls either with or without benign prostate hyperplasia (69.99 years SD ± 9.07, range 45–99) (Supplementary Data 5). Cases were relatively evenly distributed across the ISUP grade grouping, representing high (ISUP 3 to 5, 54.58%) and low-risk PCa (ISUP 1 to 2, 45.43%). As previously observed for this study population[6], PSA levels were significantly elevated, with 44.95% of cases presenting with PSA > 100 ng/ml, and only 20.85% of controls presenting with a PSA level less than the global standard for PCa diagnosis (4 ng/ml)[77].

## EWAS analysis and statistics

As the controls in this study were recruited based on a referral to a urologist clinic and were negative at histopathological examination of resected biopsy cores (on average 12 per patient), one cannot ignore the possibility of missed PCa diagnoses in these individuals. Therefore, in parallel to a classic case-control EWAS analysis (451 cases, 292 controls), we designed an additional EWAS focused on distinguishing HRPCa (ISUP 3-5, $N = 203$) versus low-risk or no PCa (LRPCa/NoPCa, $N = 461$). Cases that were missing an ISUP grading were excluded from this analysis.

RStudio v4.1.1 and PLINK v1.9 were used for analyses and visualisation[22,78]. A logistic regression using an additive genetic model accounting for age was used. A q-value false-discovery rate (FDR) cut-off of 0.05, calculated in R using the package 'qvalue'[79], was used to determine genome-wide significance. Manhattan, quantile-quantile

(QQ), and regional association plots were generated using 'ggplot2' in RStudio[73], with gene annotations for the canonical transcripts fetched through the R package 'biomaRt'[80] from Ensembl Human GRCh38.p13, version 108 and GENCODE v43 via UCSC[81]. African ancestry minor allele frequencies were fetched from gnomAD v3.1.2[67].

For the 17 out of 278 known PCa risk variants[16], the three variants from the Ghana study[10], and the previously summarised known cancer variants[13] that were available in our exome array data, the risk allele frequency in cases and controls, age-adjusted odds ratio (OR), 95% confidence intervals (CI), and P-values were calculated in PLINK v1.9[22]. For SNPs where the risk allele was the major allele in our population, the inverse odds ratio (and 95% CI) was calculated to reflect the odds of the risk allele in cases compared to controls. Three of the 17 known PCa risk SNPs were rare variants that were processed with zCall v3.4[23] (see below) to improve genotype calls. There were no changes to the allele frequencies of the three SNPs post-processing.

## Gene-based analyses with rare variants

To improve the genotype calls of rare variants (minor allele frequency (MAF) < 0.01), zCall v3.4[23] was used with a default z value of 7 (global concordance=99.19%). Quality control followed that of EWAS filtering, including strand flipping, converting coordinates from hg18 to hg38, removing triallelic or duplicated variants, and removing heterozygous chromosome X variants (Fig. 2D). A total of 31,445 rare variants (0 < MAF ≤ 0.01) remained. Processing the data with zCall reduced the total number of fixed variants from 163,838 to 162,678 SNPs (Supplementary Data 10). Variant annotations for the canonical transcripts were fetched via the R package 'biomaRt'[80] from Ensembl Human GRCh38.p13 version 108. The R package 'SKAT' version 2.2.4 was used to conduct optimal unified sequence kernel association tests (SKAT-O)[24]. SNPs were grouped by genes, and two analyses were conducted per phenotype: (1) SKAT_CommonRare to test for the combined effect of common and rare variants, using 71,908 common and rare SNPs across 15,593 genes; and (2) SKATBinary to test for rare variant associations, using 31,345 rare SNPs across 11,740 genes. The phenotypes tested were as per the EWAS analyses: PCa risk (cases vs controls) and HRPCa risk (HRPCa vs LRPCa/No PCa). The analyses were conducted chromosome by chromosome, using age as a covariate with $N = 1000$ resampling used in the null model. Significant genes were identified using the family-wise error rate (FWER) multiple testing correction (cut-off 0.05) built into the package.

## Heritability estimates

SNP-based heritability was calculated using genome-based restricted maximum likelihood (GREML) in GCTA v1.92.0[82], using autosomal variants at a prevalence of 0.001 based on the 5-year prevalence of PCa in South Africa (39,863 cases out of 29,216,012 men)[4], as well as an approximate prevalence of 0.0004 for high risk PCa based on the fraction of cases with ISUP 3-5 (43%) in this study.

## Reporting summary

Further information on research design is available in the Nature Portfolio Reporting Summary linked to this article.

# Data availability

The sequencing data analysed in this study were obtained from the European Genome-Phenome Archive (EGA; https://ega-archive.org/) under overarching accession EGAS00001006425, with access to the Southern African Prostate Cancer Study (SAPCS) Dataset (EGAD00001009067) granted by the SAPCS Data Access Committee (DAC). Exomic genotyping summary statistics have been deposited in the GWAS Catalog database (www.ebi.ac.uk/gwas) under accession code GCST90296485 for cases versus control data and GCST90296486 for high-risk PCa versus low-risk PCa and control. Polygenic risk scores are available in the PGS Catalog database

(https://www.pgscatalog.org/) under accession code PGP000516. Source data are provided with this paper. The remaining data are available within the Article, Supplementary Information or Source Data file. Source data are provided with this paper.

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

## Acknowledgements

The authors are grateful to the patients and their families who have contributed to this study, to co-founding SAPCS members the late Professor Philip A. Venter and retired urologist Dr. Richard Monare from the University of Limpopo in South Africa, as well as the Medical Research Council (MRC) of South Africa for SAPCS seed-funding; without their contribution, this research would not be possible. This study was supported by a Cancer Association of South Africa (CANSA) grant to M.S.R.B. and V.M.H., the National Health and Medical Research Council (NHMRC) of Australia via a Project grant (APP1165762) to V.M.H. and Ideas grants (APP2001098) to V.M.H. and M.S.R.B. and (APP2010551) to V.M.H., as well as partially via the U.S.A. Congressionally Directed Medical Research Programs (CDMRP) Prostate Cancer Research Program (PCRP) through an Idea Development Award (PC200390, TARGET Africa) to V.M.H. and a HEROIC Consortium Award (PC210168, HEROIC PCaPH Africa1K) to V.M.H. and M.S.R.B., acknowledging our co-leads Professors Gail S. Prins (University of Illinois at Chicago) and Mungai Peter Ngugi (University of Nairobi, Kenya). V.M.H. is further supported by the Petre Foundation via the University of Sydney Foundation, Australia.

## Author contributions

V.M.H. and M.S.R.B. co-conceived the study. V.H.H. and D.C.P. designed the experimental approach. P.X.Y.S. led the statistical analyses, with statistical and computational assistance provided by K.G., J.J. and W.J. s.v.Z., R.C. and S.B.A.M. recruited patients and provided critical clinical review. N.M., D.C.P., S.M.P. and M.S.R.B. collated the specimens and provided both quality control and administrative data support. D.C.P. prepared the DNA for analysis and quality control. P.X.Y.S. generated the

figures and provided the data analytics, with further interpretation provided by V.M.H. M.S.R.B. as the SAPCS Clinical Director, S.B.A.M. as the SAPCS Urological Director and V.M.H. as the SAPCS Scientific Director provided expertise-specific supervision. P.X.Y.S. drafted the manuscript with supervision from V.M.H., with all authors reviewing and editing the manuscript.

## Competing interests

All authors declare no competing interest.
