## [Peer Review File · Nature Communications]

Prostate cancer genetic risk and associated aggressive disease in men of African ancestryReviewers' Comments:

Reviewer #1:

Remarks to the Author:

In this paper, the authors evaluated the previous known genetic variants associated with prostate cancer in a population of 113 Black South African men with prostate cancer. They were unable to replicate previous findings of the multiethnic polygenic risk score association with prostate cancer in this population and conducted an Exome Wide Association Study and a gene-based analyses on 798 Black South African prostate cancer men. Although none of the genetic variants achieved genome-wide significant, they identified some interesting candidate SNPs.

If the manuscript is very interesting and the analyses are well conducted, I have some concerns regarding the reported data:

1) Are some of the 113 patients having undergone tumour/blood paired whole genome sequencing part of the 798 who underwent whole exome sequencing?

2) In the Result section "known risk alleles in Black South Africans with PCa", the data presented in Supplementary Tables and those provided in the text do not match. In Supplementary Table 4, 21 of the 278 risk variants were absent and 21 were fixed, leaving 236 non-fixed variants in the Black South African population whereas only 229 were reported as not fixed in the text and in Figure 1A. This is important because the polygenic risk score assessed in black South Africans included only 229 variants.

Similarly, several frequencies provided in the text for the comparisons between the South African population and African Ancestry population (Supplementary Table 4) and between the South African population and the Uganda population (Supplementary Table 5) are different from the data in the Supplementary Tables. Is the list of 3 genes with the largest differences Line 112-113 correct? The authors indicated that "18 variants were more common in Ugandan controls than in our SA population", but the difference in RAF was low for some of the variants (0.01, for example), is this difference statistically significant?

Was any of the risk allele frequency statistically different between compared populations?

3) In the Result section "Common risk variants associated with Pca in Black South Africans", the authors must indicate that 37 men with unknown age at diagnosis were excluded from the EWAS and gene-based analyses, leaving 743 included men for these analyses.

After QC, the total number of common variants should be 50,581 (Line 155, Figure 1C) instead of 50,591.

The authors reported the rs114057260 is the only predicted deleterious variant defined using/or Polyphen (Line 171-173) but in Table 1, 3 variants are predicted as deleterious, and rs56802364 is the only one with high confidence. Please verify.

The authors should indicate whether the frequencies provided (line 178) are those of the SA cases or controls and verify these frequencies with data from Supplementary Table 4.

The p-values in Lines 181-182 are also different from those in Supplementary Table 7.

4) Concerning the EWAS analysis, did the authors perform a case-case study comparing HRPCa versus Low Risk patients? Is there any significant/interesting result obtained with this analysis?

5) Is rs66883347 (CASC17) really one of the SNPs with the largest difference in RAF between South African cohort and African-ancestral controls (Line 246)? It is not in Figure 3 and it seems to be wrong based on the data in Supplementary Table 4.

6) Could the lack of replication of previous findings of the multiethnic polygenic risk score association be related to the small size of the studied population (113 men)?

7) The first paragraph of the Discussion takes up elements already presented in the introduction and must be shortened.

Minor revisions:

- Supplementary Table 1: please provide the number of patients with missing PSA level
- In Table 2, there are 4 SNPs with deleterious and potentially damaging effects based on SIFT and/or PolyPhen but only 3 are cited Line 194.
- Line 415: it should be 244,117 instead of 224,127
- The Supplementary figures are missing.

Reviewer #2:

Remarks to the Author:

Utilizing the whole-genome sequencing data in 113 Black South African men and exome-array data in 798 Black South African men, the authors compared the allele frequency of 278 previously reported prostate cancer risk variants, evaluated the association of a prostate cancer PRS with aggressive disease, and performed single-variant association and gene-based association analysis to identify variants or genes to be associated with aggressive prostate cancer in this south African population.

However, the results provided in tables does not support the findings stated in the main text in terms of large allele frequency differences observed between this SA population and in previously reported African populations (see comments below). The PRS was evaluated in 113 men that were predominately aggressive cases and only 14 of them were considered as non-aggressive cases. The definition of non-aggressive disease in this study is not clearly defined and likely to be quite different from what was used in the published study. Although the authors stated that the PRS association with aggressive PCa was not replicated in this South African population, it is my opinion that their study population was not appropriate (non-aggressive cases poorly defined) nor statistically power to evaluate the performance of PRS. In addition, the PRS association was reported as in $\log(\text{OR})$ rather in OR per standard deviation change, which is the conventional way to present PRS associations, prevented the comparison of their results to published studies.

Not surprising that no genes reached the significance in gene-based analysis given the small sample sizes.

Please see the comments below:

1. The author reported that of the 278 prostate cancer risk variants, they found 22 were absent and 27 were fixed in their southern African cohort. In Supplementary Table 4 I only found 21 variants with RAF of 0 and 21 variants with RAF of 1. The results reported in Supplementary Table 4 do not match to the results in the main text. For those variants that are absent or fixed in the SA cohort, were they absent or fixed in the previous African ancestry meta-analysis too? Are the frequencies similar?
2. It's interesting to see big differences in allele frequencies between the SA cohort and the reported meta-analysis. The authors reported 14 variants showed differences in $\text{RAF} > 0.20$. I only see 10 and none of the variants fixed in the SA population showed > 0.2 difference in RAF. It was also reported that rs34680713 showed the largest differences with $\text{RAFAA} = 0.805$ and $\text{RAFSA} = 0$. But in Supplementary Table 4, RAFSA for rs34680713 was 0.739. Similarly, a large difference in RAF was reported for rs66883347 with $\text{RAFAA} = 0.213$ and $\text{RAFSA} = 1$, while in Supplementary Table 4 RAFSA was reported as 0.128. Again, the results in Supplementary Table 4 do not match to those in the main text.
3. Since the 278 prostate cancer risk variants include some variants that are rare in African ancestry populations, should the comparison of allele frequency be limited to those that are common in African

populations? Given the small sample size of the SA population, the allele frequency estimated can be unstable and unreliable for rare variants.

4. When comparing RAF between SA population and Uganda population from published GWAS, it's better to compare to RAF reported in Ugandan cases since the SA population are all prostate cancer cases. The author reported 3 variants with $RAF > 0.15$ but the differences are sometimes based on RAF in Ugandan cases and sometimes based on RAF in Ugandan controls.

5. It was not clear to me the purpose of comparing the PRS score with age, grade group or log PSA. The score was not developed to predict age at diagnosis, or cancer grade or PSA. What does it mean if the score is correlated or not correlated?

6. It was also not clear how the PRS association was assessed. The author defined aggressive PCa but didn't mention how non-aggressive PCa were defined only that 14 samples were considered as non-aggressive cases. I am not sure how to interpret the association results in $\log(OR)$. PRS association is typically reported as OR per standard deviation (SD) change and the mean and SD should be estimated from PRS distribution of controls (non-cases). In general, I do not think this study has the proper design or the statistical power to evaluate the performance of PRS in stratifying risk of aggressive PCa.

7. It would be helpful if the risk allele frequency can be provided in Table 1 and Table 2 along with the allele frequency from public databases in African populations to show how common the variants are in SA and other AA populations. Are any of the variants in Table 1 and Table 2 except rs339331 in strong LD with previously reported risk variants ($R^2 > 0.7$ or 0.8)? Are they likely to represent novel regions?

8. Perform the gene-based analysis including both common and rare variants do not provide any additional information than the single-variant association of common variants since the association will be driven by one or more common variants associated with the outcome of interests. The only difference is that the gene-based association test has a lower significant threshold so the variants that do not reach exome-wide significance in single-variant analysis can now be considered significant. The same can be applied to genes with only one variant included in gene-based analysis.

Reviewer #3:

Remarks to the Author:

The authors have obtained remarkable results that significantly contribute to the understanding of germline genetic risk for developing prostate cancer and for high-grade disease in Black South African men. Their findings shed new light on the need to increase genomic research on prostate cancer among African men. It is noteworthy that while other studies on germline genetic markers have identified variation in Homologous recombination repair genes like BRCA in Ugandan and even in Black South African men, these genes are generally considered a risk for developing prostate cancer and utilized as biomarkers for targeted therapy, it is of interest to note that this study didn't identify such genes.

In the discussion, the authors rightly attributed the high frequency of A-allele at rs2274911 "to elevated PSA levels observed for Black South African men, irrespective of PCa status". It will be of interest if the authors could provide additional comments on the implication of this finding on the use of the global PSA standard (4 ng/ml) for Black South African men.

The methodology employed in this study is rigorous and robust. The authors have provided a detailed description of the experimental design, materials, bioinformatics and statistical analyses used. The methods used are consistent with the expected standards in the field, however, since ~80% of controls had PSA of greater than 4 ng/ml, the authors should clarify whether the pathological reports of the biopsy for some of the controls indicated the presence of Benign Prostate Hyperplasia. Indeed, understanding the alleles that could be used to distinguish between PCa and BPH would be of scientific and clinical importance.

Overall, the authors have provided sufficient detail in the methods section, enabling the work to be reproduced. The inclusion of additional details on how the controls were selected would strengthen the manuscript.

In summary, I highly recommend the publication of this manuscript after the authors have addressed these minor observations.

Response to reviewers

The authors thank the reviewers for their comments and review. All responses and changes to the Main Text of the manuscript are highlighted in red for ease of identification and each comment is addressed below.

Reviewer #1, expertise in prostate cancer genetics and genomics, GWAS and PRS

General comments: In this paper, the authors evaluated the previous known genetic variants associated with prostate cancer in a population of 113 Black South African men with prostate cancer. They were unable to replicate previous findings of the multiethnic polygenic risk score association with prostate cancer in this population and conducted an Exome Wide Association Study and a gene-based analyses on 798 Black South African prostate cancer men. Although none of the genetic variants achieved genome-wide significant, they identified some interesting candidate SNPs.

If the manuscript is very interesting and the analyses are well conducted, I have some concerns regarding the reported data:

1) Are some of the 113 patients having undergone tumour/blood paired whole genome sequencing part of the 798 who underwent whole exome sequencing?

Response: Yes, 29 samples had both been genotyped on the Exome array and underwent tumour/blood whole genome sequencing. Due to the technologies, there were differences in variant calls, with up between 1201 to 1395 variants (out of 243,051 SNPs) in a single individual with differing genotype calls. As it would be difficult to rectify the correct variant calls, we treated the datasets individually.

2) In the Result section “known risk alleles in Black South Africans with PCa”, the data presented in Supplementary Tables and those provided in the text do not match. In Supplementary Table 4, 21 of the 278 risk variants were absent and 21 were fixed, leaving 236 non-fixed variants in the Black South African population whereas only 229 were reported as not fixed in the text and in Figure 1A. This is important because the polygenic risk score assessed in black South Africans included only 229 variants.

Response: Thank you for picking up on this error. We have checked the list of risk variants and have corrected the table and the manuscript. Upon closer inspection and verifying risk/reference alleles using the hs38DH.fasta file, 18 variants were not found (fixed for the reference allele), 21 variants were fixed for the risk allele, and 8 variants were excluded due to repeat regions or indels where the risk allele was not clear. This left 231 variants for polygenic risk scoring – the analysis was redone. Supplementary Table 4 (now Supplementary Table 2) has been updated to reflect this and a column has been added for easier identification of the variants used in the polygenic risk scoring. The Figure 2 summary slide has also been updated.

Similarly, several frequencies provided in the text for the comparisons between the South African population and African Ancestry population (Supplementary Table 4) and between the South African population and the Uganda population (Supplementary Table 5) are

different from the data in the Supplementary Tables. Is the list of 3 genes with the largest differences Line 112-113 correct?

Response: We have updated these allele frequencies. The largest differences have now changed to (Line 112): “The largest differences included rs111595856 (*INHBB*, $RAF_{SA} = 0.67$, $RAF_{AA} = 0.398$), rs35159226 (*ZNF322*, $RAF_{SA} = 0.562$, $RAF_{AA} = 0.308$), and rs8005621 (*SALRNA1*, $RAF_{SA} = 0.549$, $RAF_{AA} = 0.2$).” The text for the comparison between Ugandan cases and controls to South African cases has been updated for clarity (Line 116): “Among the top 136 associated variants in the Ugandan GWAS, three variants showed differences in $RAF > 0.15$ compared to Ugandan cases and controls, rs6431219 (*BIN1*, $RAF_{SA} = 0.416$, $RAF_{UGPCS_Cases} = 0.6$, $RAF_{UGPCS_Controls} = 0.51$), rs61005944 (*ENSG00000237101*, $RAF_{SA} = 0.527$, $RAF_{UGPCS_Cases} = 0.31$, $RAF_{UGPCS_Controls} = 0.22$) and rs140698498 (*RBFox1*, $RAF_{SA} = 1$, $RAF_{UGPCS_Cases} = 0.87$, $RAF_{UGPCS_Controls} = 0.8$)”

The authors indicated that “18 variants were more common in Ugandan controls than in our SA population”, but the difference in RAF was low for some of the variants (0.01, for example), is this difference statistically significant?

Response: Statistical significance for this difference was not investigated. If Ugandan and South African ancestries shared genetic risk profiles for prostate cancer, we would expect a higher frequency of these alleles that increase prostate cancer risk in South African cases than Ugandan controls, thus we wanted to highlight that the frequency of risk alleles was greater in Ugandan controls than South African cases. Small differences in RAF between Ugandan controls and South African cases are also indicative that this set of variants do not explain the South African PCa risk well.

Was any of the risk allele frequency statistically different between compared populations?

Response: This was not compared. Given that our sample size was small (113 individuals), our RAF may not be very precise so comparisons to much larger populations (such as the AAPC) may not be accurate.

3) In the Result section “Common risk variants associated with Pca in Black South Africans”, the authors must indicate that 37 men with unknown age at diagnosis were excluded from the EWAS and gene-based analyses, leaving 743 included men for these analyses.

Response: A sentence has been added at line 165, which reads: “For exome-wide association analysis (EWAS) and gene-based analysis, a total of 37 men with unknown age at diagnosis were excluded, leaving 743 men.”

After QC, the total number of common variants should be 50,581 (Line 155, Figure 1C) instead of 50,591.

Response: Sorry, the authors could not find anywhere in the document it states 50,581 variants. 50,591 is the correct number and this is on line 155 (now line 168), in the methods, and Figure 1C.

The authors reported the rs114057260 is the only predicted deleterious variant defined using/or Polyphen (Line 171-173) but in Table 1, 3 variants are predicted as deleterious, and rs56802364 is the only one with high confidence. Please verify.

Response: For clarity, predicted deleterious variants were those defined as being deleterious/damaging by both SIFT and PolyPhen. The other two variants rs34759333 and rs56802364 were considered deleterious by SIFT but benign by PolyPhen, so these were not classed as a predicted deleterious variant. Line 185 has now been updated for clarity: "...in *ZZEF1* which is the only predicted deleterious variant (PDV), defined as variants predicted to be deleterious by SIFT and damaging (or possibly damaging) by PolyPhen."

The authors should indicate whether the frequencies provided (line 178) are those of the SA cases or controls and verify these frequencies with data from Supplementary Table 4. The p-values in Lines 181-182 are also different from those in Supplementary Table 7.

Response: Line 190 has been updated to clarify that the minor allele frequencies were a population-level frequency for the 743 individuals: "Among the remaining SNPs, across the whole population of 743 individuals three are rare (MAF<0.01), while none of the remaining nine SNPs (MAF 0.015 to 0.49) showed risk association (all P>0.25) (**Figure 4A**)."

P-values have now been fixed to reflect those correctly indicated in the Supplementary Table 7 (now Supplementary Table 6 Line 194-197): "Among these, the top associated variants included the *RFX6* SNP rs339331 (P=0.0002), intergenic variant rs9600079 (pseudogene *RNU4-10P* 37.5kb downstream, closest protein-coding gene is *KLF5* 76kb upstream, P=0.0014), and *CASC8/PCAT1* variant rs445114 (P=0.0047)."

4) Concerning the EWAS analysis, did the authors perform a case-case study comparing HRPCa versus Low Risk patients? Is there any significant/interesting result obtained with this analysis?

Response: Due to limitations in sample size we chose to compare HRPCa against LRPCa and controls together, rather than HRPCa vs LRPCa alone.

5) Is rs66883347 (*CASC17*) really one of the SNPs with the largest difference in RAF between South African cohort and African-ancestral controls (Line 246)? It is not in Figure 3 and it seems to be wrong based on the data in Supplementary Table 4.

Response: This has now been updated to reflect the correct analysis (Line 263): ", large differences in RAF (>0.15) were observed for 11 variants, including those in the genes *INHBB* (rs111595856), *ZNF322* (rs35159226), *SALRNA1* (rs8005621), *FGF10* (rs1482675), *HNF1B* (rs11263763), *PCAT19* (rs11673591), and *TAB3* (rs5972255)."

6) Could the lack of replication of previous findings of the multiethnic polygenic risk score association be related to the small size of the studied population (113 men)?

Response: If the variants utilised in a multiethnic polygenic risk score captures all (or most of) the genetic variation observed between men of all ethnicities with aggressive PCa from those without, we should still see a difference in our population even though the sample size is small. Considering the results of our analysis showed no difference in the mean polygenic scores between aggressive and non-aggressive disease by both definitions, this suggests that the genetic variation associated with South African aggressive PCa is not well captured by these 278 risk variants. These findings will certainly be clearer in a larger cohort, hence our team is dedicated to increasing the availability of African-relevant data, which is significantly lacking.

7) The first paragraph of the Discussion takes up elements already presented in the introduction and must be shortened.

Response: We thank you for the feedback - the first paragraph of the Discussion has now been shortened as requested (Line 249): “In this study, motivated by limited studies having identified three African-specific protein altering risk alleles^{19, 25}, we examined PCa risk and aggressive disease associations and have provided much needed evaluation of known PCa risk alleles within the under-represented region of southern Africa. While dwarfed in sample size compared to European-ancestral or African American GWAS studies, this study highlights significant resources and efforts needed to elucidate the genetic contribution to ancestrally-driven PCa health disparities across the African diaspora. “

Minor revisions:

1. Supplementary Table 1: please provide the number of patients with missing PSA level.

Response: This has been added to the table and updated to reflect Jaratlerdsiri et al., 2022 Nature (DOI: 10.1038/s41586-022-05154-6), from which the sequencing data was sourced.

2. In Table 2, there are 4 SNPs with deleterious and potentially damaging effects based on SIFT and/or PolyPhen but only 3 are cited Line 194.

Response: This line has been updated to include the 4th SNP (Line 209): “SNPs rs60322991 (*ABCB6*), rs877834 (*NPVF*), rs2075662 (*COL15A1*), and rs77944357 (*ABCA10*) have deleterious and potentially damaging effects based on SIFT and/or PolyPhen yet are benign or are lacking prediction by ClinVar (Table 2).”

3. Line 415: it should be 244,117 instead of 224,127.

Response: This line (Line 424) has been updated to 244,217 (247,870 in the assay minus 3653 with poor GenTrain score and poor call frequency)

4. The Supplementary figures are missing.

Response: All documents have been checked that they are accurately uploaded.

Reviewer #2, expertise in prostate cancer genetics and genomics, GWAS, PRS, statistics and epidemiology and African ancestries

General comments: Utilizing the whole-genome sequencing data in 113 Black South African men and exome-array data in 798 Black South African men, the authors compared the allele frequency of 278 previously reported prostate cancer risk variants, evaluated the association of a prostate cancer PRS with aggressive disease, and performed single-variant association and gene-based association analysis to identify variants or genes to be associated with aggressive prostate cancer in this south African population.

However, the results provided in tables does not support the findings stated in the main text in terms of large allele frequency differences observed between this SA population and in

previously reported African populations (see comments below). The PRS was evaluated in 113 men that were predominately aggressive cases and only 14 of them were considered as non-aggressive cases. The definition of non-aggressive disease in this study is not clearly defined and likely to be quite different from what was used in the published study. Although the authors stated that the PRS association with aggressive PCa was not replicated in this South African population, it is my opinion that their study population was not appropriate (non-aggressive cases poorly defined) nor statistically power to evaluate the performance of PRS. In addition, the PRS association was reported as in log(OR) rather in OR per standard deviation change, which is the conventional way to present PRS associations, prevented the comparison of their results to published studies.

Response: Thank you for your feedback. We have now added for clarity the definition of aggressive/non-aggressive disease in Figure 2 as well as in the text. Upon reviewing the supplementary tables, we noticed an error in the phenotypes of the samples (Supplementary Table 1) and for the allele frequencies for the 278 risk variants (Supplementary Table 4 – now Supplementary Table 2).

Supplementary Table 1 has now been updated to be consistent with Jaratlerdsiri et al., 2022 Nature (DOI: 10.1038/s41586-022-05154-6), from which the sequencing data was sourced.

Supplementary Table 4 (now Supplementary Table 2) has been updated for greater clarity, including the vcf IDs and clearly states the reference and alternate alleles for these variants in our data (particularly important for indel repeat regions where the risk allele is unclear). We have also included a column stating which variants were included in the polygenic risk scoring and have put in detailed comments for inclusions/exclusions.

PRS associations. We concur with your suggestion to report the PRS associations. We have now added OR per standard deviation to our results.

Statistical power. We acknowledge the limited statistical power in our study, however given the scarcity of research and genetic data within Sub-Saharan Africa, our study is an important first steppingstone in addressing ethnic disparity not only in study participation but in prostate cancer outcomes. Our samples, while limited, are the only ones currently available to evaluate how polygenic risk scores based on primarily African American or European ethnicities apply to Southern Africa. Due to limited non-aggressive cases, comparisons between deciles or quartiles were not done, but this should certainly be validated in future studies with larger sample sizes. Given the extremely limited genetic data in Sub-Saharan Africa

Not surprising that no genes reached the significance in gene-based analysis given the small sample sizes.

Response: While the authors agree on the small sample size, we will strongly argue that we need to begin to address prostate cancer risk in African populations and accept that this is a new and only now developing field of research. Our study brings awareness with regards to the lack of power and lack of focused research within Africa. However, we did detect significant and relevant genes in the gene-based analysis, including *H3C1*, *MTG1*, and *MBP* in the rare variant PCa analysis, and *KLF5* in the common and rare variant HRPCa analysis (refer to lines 329 to 352 in the Discussion). It is unsurprising that our EWAS results did not reach genome-wide significance given our sample size. Despite the limited size we can still discover variants and genomic regions that are the most associated with PCa (thus we reported the

top associations with a relaxed cutoff of $P < 5 \times 10^{-4}$), except with a higher frequency of false positives. While it will need validation in the future, we still believe there is great value in publishing this analysis as a first step to overcoming barriers in globally inclusive genomics and prostate cancer research.

Please see the comments below:

1. The author reported that of the 278 prostate cancer risk variants, they found 22 were absent and 27 were fixed in their southern African cohort. In Supplementary Table 4 I only found 21 variants with RAF of 0 and 21 variants with RAF of 1. The results reported in Supplementary Table 4 do not match to the results in the main text. For those variants that are absent or fixed in the SA cohort, were they absent or fixed in the previous African ancestry meta-analysis too? Are the frequencies similar?

Response: As addressed and also noted by Reviewer 1, we have now fixed Supplementary Table 4 (now Supplementary Table 2) as stated above: 18 variants were not found (fixed for the reference allele), 21 variants were fixed for the risk allele, and 8 variants were excluded due to repeat regions or indels where the risk allele was not clear, leaving 231 variants for polygenic risk scoring. We have added a column in Supplementary Table 2 showing the differences between RAF for the SA cohort and the African ancestry cohort for clarity. The difference in RAF between the populations for the variants fixed in the SA cohort ranged from -0.117 to 0.045.

2. It's interesting to see big differences in allele frequencies between the SA cohort and the reported meta-analysis. The authors reported 14 variants showed differences in $RAF > 0.20$. I only see 10 and none of the variants fixed in the SA population showed > 0.2 difference in RAF. It was also reported that rs34680713 showed the largest differences with $RAF_{AA} = 0.805$ and $RAF_{SA} = 0$. But in Supplementary Table 4, RAF_{SA} for rs34680713 was 0.739. Similarly, a large difference in RAF was reported for rs66883347 with $RAF_{AA} = 0.213$ and $RAF_{SA} = 1$, while in Supplementary Table 4 RAF_{SA} was reported as 0.128. Again, the results in Supplementary Table 4 do not match to those in the main text.

Response: We have updated the plot as per the changes in Supplementary Table 4 (now Supplementary Table 2). This paragraph has now been updated to read (Line 109-114): "When compared to previously published risk allele frequencies (RAF) for African-ancestral (AA) controls¹⁶, 11 showed differences in $RAF > 0.15$, of which eight were more common in our SA population and three were more common in the published largely US-derived AA data (Figure 3, Supplementary Table 2). The largest differences included rs111595856 (*INHBB*, $RAF_{SA} = 0.67$, $RAF_{AA} = 0.398$), rs35159226 (*ZNF322*, $RAF_{SA} = 0.562$, $RAF_{AA} = 0.308$), and rs8005621 (*SALRNA1*, $RAF_{SA} = 0.549$, $RAF_{AA} = 0.2$)."

3. Since the 278 prostate cancer risk variants include some variants that are rare in African ancestry populations, should the comparison of allele frequency be limited to those that are common in African populations? Given the small sample size of the SA population, the allele frequency estimated can be unstable and unreliable for rare variants.

Response: While it is certainly true that the allele frequency may not be precise given our small population, conversely, given there is limited African genetic data we see the immense

value in reporting the frequency of all 278 variants and comparing them to African ancestry frequencies. Supplementary Table 2 will be useful for people interested in the specifics.

4. When comparing RAF between SA population and Uganda population from published GWAS, it's better to compare to RAF reported in Ugandan cases since the SA population are all prostate cancer cases. The author reported 3 variants with $RAF > 0.15$ but the differences are sometimes based on RAF in Ugandan cases and sometimes based on RAF in Ugandan controls.

Response: This has now been edited to include both cases and controls for RAF for clarity (Line 114 – 119): “Among the top 136 associated variants in the Ugandan GWAS, three variants showed differences in $RAF > 0.15$ compared to Ugandan cases and controls, rs6431219 (*BIN1*, $RAF_{SA} = 0.416$, $RAF_{UGPCS_Cases} = 0.6$, $RAF_{UGPCS_Controls} = 0.51$), rs61005944 (*ENSG00000237101*, $RAF_{SA} = 0.527$, $RAF_{UGPCS_Cases} = 0.31$, $RAF_{UGPCS_Controls} = 0.22$) and rs140698498 (*RBFox1*, $RAF_{SA} = 1$, $RAF_{UGPCS_Cases} = 0.87$, $RAF_{UGPCS_Controls} = 0.8$) (Supplementary Figure 1).”

5. It was not clear to me the purpose of comparing the PRS score with age, grade group or log PSA. The score was not developed to predict age at diagnosis, or cancer grade or PSA. What does it mean if the score is correlated or not correlated?

Response: Agreed. This was originally done as grade group and PSA are used to classify individuals into aggressive/non-aggressive disease. This has now been removed and replaced with a histogram of scores for each group.

6. It was also not clear how the PRS association was assessed. The author defined aggressive PCa but didn't mention how non-aggressive PCa were defined only that 14 samples were considered as non-aggressive cases. I am not sure how to interpret the association results in log(OR). PRS association is typically reported as OR per standard deviation (SD) change and the mean and SD should be estimated from PRS distribution of controls (non-cases). In general, I do not think this study has the proper design or the statistical power to evaluate the performance of PRS in stratifying risk of aggressive PCa.

Response: We have now clarified the definition of aggressive/non-aggressive disease in Figure 2 as well as in the text (Line 126-131): “The PRS was evaluated using two definitions of aggressive PCa, firstly Chen et al., 2023's definition: ISUP 4-5 or $PSA \geq 20ng/ml$, which grouped our samples into $N=101$ aggressive and $N=11$ non-aggressive (one sample with missing PSA and ISUP excluded); and our definition: ISUP 3-5, grouping our samples into $N=87$ aggressive and $N=18$ non-aggressive (eight samples with missing ISUP excluded) (Figure 2).”

We have also added in mean, SD and range of scores for each group (Line 133-142): “Using Chen et al., 2023's definition of aggressiveness, for African weights, for the aggressive group the mean score was 0.034 (SD = 0.002, range 0.029 – 0.039) while that of the non-aggressive group was 0.034 (SD = 0.002, range 0.032 – 0.037). For multiethnic weights, the aggressive group's mean score was 0.041 (SD = 0.002, range 0.035 – 0.046), and non-aggressive was 0.041 (SD = 0.002, range 0.039 – 0.045). Using our definition of aggressiveness, for African weights, the aggressive group had a mean score of 0.034 (SD = 0.002, range 0.029 – 0.039), while the non-aggressive group had a mean score of 0.034 (SD = 0.002, range 0.032 – 0.037). With multiethnic weights, we observed for the aggressive group a mean 0.041 (SD = 0.02,

range 0.035 – 0.046) and for the non-aggressive group a mean 0.041 (SD=0.02, range 0.039 – 0.045).“

As mentioned above we have now added OR per standard deviation change (Line 144-147): “African score OR per SD = 1.38, 95% CI = 0.72 to 2.66; multiethnic score OR per SD = 1.24, 95% CI = 0.65 to 2.34; nor using our definition of HRPCa: African score OR per SD = 1.01, 95% CI = 0.6 to 1.71; multiethnic score OR per SD = 1.13, 95% CI = 0.67 to 1.9.”

7. It would be helpful if the risk allele frequency can be provided in Table 1 and Table 2 along with the allele frequency from public databases in African populations to show how common the variants are in SA and other AA populations. Are any of the variants in Table 1 and Table 2 except rs339331 in strong LD with previously reported risk variants ($R^2 > 0.7$ or 0.8)? Are they likely to represent novel regions?

Response: Complying with your suggestion, we have added the minor allele frequencies from our data and African ancestry from gnomAD v3.1.2 into the tables. Additionally, we found no strong LD between our top variants and previously reported cancer risk variants (397 available on the array out of 2477) from Harlemon et al., 2020 (Supplementary Table 7 – now Supplementary Table 6). There may be other risk variants that are in linkage, however, they would not be detected due to how the genotype array was designed and which variants were selected for the array.

8. Perform the gene-based analysis including both common and rare variants do not provide any additional information than the single-variant association of common variants since the association will be driven by one or more common variants associated with the outcome of interests. The only difference is that the gene-based association test has a lower significant threshold so the variants that do not reach exome-wide significance in single-variant analysis can now be considered significant. The same can be applied to genes with only one variant included in gene-based analysis.

Response: Given our small sample size for GWAS, we do need a lower significance threshold, and we have found value in the analysis as it picked up on relevant genes, such as *KLFS* in the HRPCa analysis including common and rare variants, in which 1 rare and 3 common variants were in (Supplementary Figure 16).

Reviewer #3, expertise in cancer genetics and genomics, prostate cancer, cancers in Africa and health disparities

General comments: The authors have obtained remarkable results that significantly contribute to the understanding of germline genetic risk for developing prostate cancer and for high-grade disease in Black South African men. Their findings shed new light on the need to increase genomic research on prostate cancer among African men. It is noteworthy that while other studies on germline genetic markers have identified variation in Homologous recombination repair genes like BRCA in Ugandan and even in Black South African men, these genes are generally considered a risk for developing prostate cancer and utilized as biomarkers for targeted therapy, it is of interest to note that this study didn't identify such genes.

Response: We thank the reviewer for their positive observations and understanding for the need for genomic-based research focused on African populations.

In the discussion, the authors rightly attributed the high frequency of A-allele at rs2274911 “to elevated PSA levels observed for Black South African men, irrespective of PCa status”. It will be of interest if the authors could provide additional comments on the implication of this finding on the use of the global PSA standard (4 ng/ml) for Black South African men.

Response: It is known that healthy men of African ancestry have PSA levels often higher than the global standard, this has been commented on in several previous studies, for example:

- Shenoy, D., Packianathan, S., Chen, A.M. et al. Do African-American men need separate prostate cancer screening guidelines?. *BMC Urol* 16, 19 (2016). <https://doi.org/10.1186/s12894-016-0137-7>
- Barlow, M., Down, L., Mounce, L.T.A. et al. Ethnic differences in prostate-specific antigen levels in men without prostate cancer: a systematic review. *Prostate Cancer Prostatic Dis* 26, 249–256 (2023). <https://doi.org/10.1038/s41391-022-00613-7>

This was also examined in our SAPCS cohort previously, as cited on line 106.

- Tindall, E.A., Monare, L.R., Petersen, D.C. et al. Clinical presentation of prostate cancer in Black South Africans. *The Prostate* 74, 8, 880-891 (2014). <https://doi.org/10.1002/pros.22806>

As other studies have a much more in-depth look at the global PSA standard, we feel it is not necessary to comment on in this paper as it was not the focus of our study. The authors are however addressing this vital observation in a larger study underway.

The methodology employed in this study is rigorous and robust. The authors have provided a detailed description of the experimental design, materials, bioinformatics and statistical analyses used. The methods used are consistent with the expected standards in the field, however, since ~80% of controls had PSA of greater than 4 ng/ml, the authors should clarify whether the pathological reports of the biopsy for some of the controls indicated the presence of Benign Prostate Hyperplasia. Indeed, understanding the alleles that could be used to distinguish between PCa and BPH would be of scientific and clinical importance.

Response: Many of the controls did present with BPH at biopsy. While the sequenced samples were all checked by a single pathologist, it was not possible to perform pathology rescoring for the complete EWAS cohort and as such we would be uncertain if no BPH was indicated or if this data was simply not provided. To avoid spurious results, we elected not to address presence of BPH.

Overall, the authors have provided sufficient detail in the methods section, enabling the work to be reproduced. The inclusion of additional details on how the controls were selected would strengthen the manuscript.

Response: Controls were selected based on, no “**histopathological**” evidence of cancer (**i.e. no Gleason score**, N=292), and the sentence adjusted as such under Results (line 164). In the methods (line 462-463), further clarification is provided that controls presented “either with or without benign prostatic hyperplasia.”

In summary, I highly recommend the publication of this manuscript after the authors have addressed these minor observations.

Response: We thank the reviewer for his/her support.

Reviewers' Comments:

Reviewer #1:

Remarks to the Author:

The authors responded to the main concerns of the reviewers, but there are minor concerns that still need to be addressed:

- Line 114: the value of the RAFAA should be 0.348 and not 0.2 (according to Supplementary Table 2)
- Lines 119-120: according to Supplementary Table 3, 19 variants (instead of 18) were more common in Ugandan controls than in the SA population (difference in RAF ranged from 0.01 to 0.094, instead of 0.01 to 0.307).
- Line 123: according to Supplementary Table 4, a total of 20 (instead of 19) of the variants were more common in Ghanaian controls than in SA cases.
- Line 168, Line 433 and Figure 2C: the total of excluded SNPs is 192,606. This leaves 50,681 of the 247,870 SNPs after the QC (and not the reported 50,591 SNPs).
- Line 191: across the whole population of 743 individuals, it seems that there are one rare SNP and two very frequent SNPs (not 3 rare ones). Moreover, the MAF of the remaining nine SNPs ranged from 0.07 to 0.99 (instead of 0.015 to 0.49).
- Please provide the name of all the SNPs that are in strong or moderate linkage disequilibrium on the Supplementary Figures 9, 10 and 12.

Reviewer #2:

Remarks to the Author:

General Comments: I appreciate the authors' efforts to address most of my comments. I only have one remaining question regarding the PRS.

The PRS distribution (mean and SD) reported in this study appeared to be too small (e.g. 0.02 to 0.04). Given that > 200 variants were included in the calculation and the weights were mostly positive (for the risk allele) and was large for some variants (e.g. weight = 0.73 for rs72725854 at 8q24 region), the PRS (if calculated as the sum of the product of dosage and weight) are more likely to be in a range of 10-30 rather than 0.02-0.04. The variant rs72725854 itself accounts for 0.73 of the PRS for men with only 1 copy of the risk allele and 1.46 of the PRS for men with 2 copies of the risk allele. Not to mention that 230 additional variants were added to the PRS. Some clarifications are needed to help understand the PRS construction and its associations.

Speaking of rs72725854, an African-ancestry specific variant, it was the variant showed the strongest association with PCa risk and accounted for the largest proportion of familiar risk of PCa in African ancestry populations. It is interesting to see that the frequency of this variant more than doubled in this SA population (13.7% vs. 6.1%, Supplementary Table 2). May be worth mentioning in the manuscript.

Reviewer #3:

Remarks to the Author:

The authors have adequately clarified the concerns that I raised and have provided acceptable responses to the questions.

REVIEWER COMMENTS

Changes in Main Text (Revision 2) are made in **RED**

Reviewer #1 (Remarks to the Author):

The authors responded to the main concerns of the reviewers, but there are minor concerns that still need to be addressed:

- Line 114: the value of the RAFAA should be 0.348 and not 0.2 (according to Supplementary Table 2)

Response: This has been corrected in the main text as $RAF_{AA} = 0.348$.

- Lines 119-120: according to Supplementary Table 3, 19 variants (instead of 18) were more common in Ugandan controls than in the SA population (difference in RAF ranged from 0.01 to 0.094, instead of 0.01 to 0.307).

Response: The line has been updated to “A total of 19 variants were more common in Ugandan controls than in our SA population”. The difference in RAF has been amended to 0.001 to 0.307.

- Line 123: according to Supplementary Table 4, a total of 20 (instead of 19) of the variants were more common un Ghanaian controls than in SA cases.

Response: We have made no changes to the sentence as we still observe 19 of the variants to be more common in the Ghanaian controls, while a single variant rs7090925 have the same RAF for the Ghanaian controls and SA cases with a $RAF=0.159$.

- Line 168, Line 433 and Figure 2C: the total of excluded SNPs is 192,606. This leaves 50,681 of the 247,870 SNPs after the QC (and not the reported 50,591 SNPs).

Response: 50,591 is the correct number after QC. However, we did note an error in the number of variants after exporting from GenomeStudio to PLINK (meant to be 244,127 instead of 244,217) – this is now updated for lines 425-426 and in Figure 2.

- Line 191: across the whole population of 743 individuals, it seems that there are one rare SNP and two very frequent SNPs (not 3 rare ones). Moreover, the MAF of the remaining nine SNPs ranged from 0.07 to 0.99 (instead of 0.015 to 0.49).

Response: For clarity, the sentence has been updated to read as follows: “Only 17 of the 278 known PCa risk variants¹⁶ were captured by the exome array data (Supplementary Table 2), with three SNPs found to be fixed for the risk allele (rs77482050, rs33984059, rs61752561), an additional two almost fixed (rs138708, rs17804499), two were fixed for the reference allele (rs77559646, rs74911261), and one rare ($MAF<0.01$ rs76832527) in our SA study population. None of the nine remaining SNPs (MAF 0.015 to 0.49) showed risk association (all $P>0.25$) (Figure 4A).”

- Please provide the name of all the SNPs that are in strong or moderate linkage disequilibrium on the Supplementary Figures 9, 10 and 12.

Response: A line has been added to the end of the figure legends for clarity, including **Supplementary Figure 9**, which reads: “Variants in strong to moderate linkage to rs339331 include rs2274911 ($r^2=0.95$) and rs636252 ($r^2=0.54$).” and for **Supplementary Figure 12**: “The variant in strong linkage to rs8473 is rs1063535 ($r^2=0.92$), while variants in moderate linkage include rs34750407, rs11016071, rs10082391, rs1050767, rs12777740, rs11016076, and rs7095325 ($r^2=0.4$ to 0.64).” For **Supplementary Figure 10**, there is no change since there are no moderate to strongly linked SNPs to rs7963300. No changes have been made to the main text.

Reviewer #2 (Remarks to the Author):

General Comments: I appreciate the authors’ efforts to address most of my comments. I only have one remaining question regarding the PRS.

The PRS distribution (mean and SD) reported in this study appeared to be too small (e.g. 0.02 to 0.04). Given that > 200 variants were included in the calculation and the weights were mostly positive (for the risk allele) and was large for some variants (e.g. weight = 0.73 for rs72725854 at 8q24 region), the PRS (if calculated as the sum of the product of dosage and weight) are more likely to be in a range of 10-30 rather than 0.02-0.04. The variant rs72725854 itself accounts for 0.73 of the PRS for men with only 1 copy of the risk allele and 1.46 of the PRS for men with 2 copies of the risk allele. Not to mention that 230 additional variants were added to the PRS. Some clarifications are needed to help understand the PRS construction and its associations.

Response: As mentioned in the methods, the PRS was scored using PLINK. As per PLINK’s documentation, the default score is calculated using the sum of the product of dosage and weight, and divided by the total number of alleles. Since 231 variants were used, the range mentioned can be achieved by multiplying our scores by 231*2 alleles. For clarification, the following sentence has been added to the methods from line 410: “We scored the South African cases via PLINK v1.9²² using default settings based on their genotypes at 231 out of 278 available risk variants (**Figure 2A**) using multiethnic and African ancestry weights¹⁶”

Speaking of rs72725854, an African-ancestry specific variant, it was the variant showed the strongest association with PCa risk and accounted for the largest proportion of familiar risk of PCa in African ancestry populations. It is interesting to see that the frequency of this variant more than doubled in this SA population (13.7% vs. 6.1%, Supplementary Table 2). May be worth mentioning in the manuscript.

Response: We thank you for bringing this interesting finding to our attention. It has now been added to the discussion and referenced, which from line 263 now reads: “Notably, the variant rs72725854, which is the most strongly associated risk variant for PCa in men of African ancestry with an allele frequency of 6.1% (Walavalkar et al., 2020), more than doubled in frequency in our South African population (13.7%).”

Reviewer #3 (Remarks to the Author):

The authors have adequately clarified the concerns that I raised and have provided acceptable responses to the questions.

Response: Appreciated.

Reviewers' Comments:

Reviewer #2:

Remarks to the Author:

I don't have any further comments. But I want to correct one comment I made about rs72725854. The higher frequency of this variant in this South African population is because it is a case-only population. The MAF of this variant in this South African population is actually quite similar to those reported in PCa cases in AAPC study and in UGPCS. I would suggest not to include the statement in line 263-265.

Reviewers Comment – Response

Reviewer #2 (Remarks to the Author):

I don't have any further comments. But I want to correct one comment I made about rs72725854. The higher frequency of this variant in this South African population is because it is a case-only population. The MAF of this variant in this South African population is actually quite similar to those reported in PCa cases in AAPC study and in UGPCS. I would suggest not to include the statement in line 263-265.

Response: While we left the reference, we adjusted the sentence emphasise our case-only analyses and reads as follows, “Notably, the variant rs72725854, which is the most strongly associated risk variant for PCa in men of African ancestry with an allele frequency of 6.1%²⁶, was present at a frequency of 13.7% in our case-only South African population.”